# Vitamin D and Acute Kidney Injury: A Reciprocal Relationship

**DOI:** 10.3390/biom15040586

**Published:** 2025-04-15

**Authors:** Chandrashekar Annamalai, Pragasam Viswanathan

**Affiliations:** Renal Research Lab, Pearl Research Park, School of Biosciences and Technology, VIT, Vellore 632014, Tamil Nadu, India; dr_a_chandrashekar@hotmail.com

**Keywords:** vitamin D, 25-hydroxyvitamin D, 1,25-dihydroxyvitamin D, calcitriol, hypovitaminosis D, hypervitaminosis D, acute kidney injury

## Abstract

Vitamin D is a sterol prohormone with no intrinsic biological activity. Calcitriol, the active form of vitamin D, is synthesized in the kidneys. It has well-known pleiotropic and cytoprotective properties. In addition to regulating parathyroid hormone secretion and enhancing gut calcium absorption, it exhibits antioxidant, anti-inflammatory, antiproliferative, and antineoplastic effects. However, the role of vitamin D in AKI is unclear, unlike in CKD. Thus, this review aimed to understand how dysregulated vitamin D homeostasis occurs in AKI, as well as to explore how vitamin D deficiency and excess influence AKI. A comprehensive literature search was conducted between January 2000 and June 2024 to uncover relevant works detailing vitamin D homeostasis in health as well as investigating the impact of vitamin D deficiency and excess in humans, animals, and in vitro cell models of AKI. According to the findings of this review, vitamin D appears to have a reciprocal relationship with AKI. Acute renal injury, among other factors, can cause hypo- or hypervitaminosis D. Conversely, AKI can also be caused by vitamin D deficiency and toxicity. Even though hypovitaminosis D is associated with AKI, it is uncertain how it impacts AKI outcomes in distinct clinical scenarios. Newer therapeutic options might emerge as a result of understanding these challenges. Vitamin D supplementation may ameliorate renal injury but needs further validation. Furthermore, hypervitaminosis D has also been implicated in AKI by causing hypercalcemia and hyperphosphatemia. It is crucial to avoid prolonged, uncontrolled, and unsupervised supraphysiological vitamin D administration, especially intramuscular injection.

## 1. Introduction

Vitamin D is a prohormone that has no intrinsic biological activity and can be obtained both endogenously from the skin and exogenously from foods and supplements [1]. The side chains of vitamin D2 (ergocalciferol) and vitamin D3 (cholecalciferol) differ, affecting their ability to bind to vitamin D-binding protein (DBP) and their efficacy. Vitamin D3 is far more effective than vitamin D2. UV rays (270 to 300 nm) photolytically convert 7-dehydrocholesterol by breaking its B ring to generate pre-vitamin D3, which then undergoes thermal isomerization to vitamin D3 after exposure to the sun. Vitamin D3 bound to DBP is hydroxylated first in the liver by cytochrome P450s (microsomal *CYP2R1* and mitochondrial *CYP27A1*) [2] to form 25-hydroxyvitamin D3 and then in the proximal tubules of the kidney by 1-hydroxylase (*CYP27B1*) to form bioactive 1,25-dihydroxyvitamin D3, also known as calcitriol [3].

The pleiotropic effects of calcitriol are well established. Apart from modulating parathyroid hormone secretion and increasing gut calcium absorption, it has also been shown to have antioxidant, anti-inflammatory, antiproliferative, and antineoplastic properties [4,5]. Calcitriol is cytoprotective by nature due to these attributes [6]. Furthermore, unlike in chronic kidney disease (CKD) [7], the role of vitamin D in AKI is not well understood. Therefore, this review sought to understand how both hypovitaminosis D and hypervitaminosis D affect clinical outcomes in AKI, as well as whether vitamin D supplementation could help prevent renal injury.

## 2. Methodology

This review sought to address several pertinent issues: (a) to discuss vitamin D homeostasis in health; (b) to understand how dysregulated vitamin D metabolism occurs in AKI; (c) to evaluate the influence of AKI on vitamin D deficiency and vitamin D excess, and (d) to review the impact of vitamin D deficiency as well as vitamin D excess on AKI.

### 2.1. Eligibility Criteria

In vitro cell-based renal models, animal models of AKI, and human participants aged ≥18 years and ≤65 years belonging to either gender, who had developed AKI due to various causes and were also vitamin D deficient or toxic, were all part of this study. Papers were excluded if they did not fit within the conceptual framework of this study, such as those focusing on the pediatric age group (<18 years) or elderly patients (>65 years), and those with chronic kidney disease or who underwent renal transplantation.

### 2.2. Information Sources

Potentially relevant documents, including peer-reviewed journal papers, systematic reviews, meta-analyses, e-books, theses, dissertations, letters, guidelines, websites, blogs, and conference materials of high esteem, written in English were identified by searching the following bibliographic databases from January 2000 to June 2024: MEDLINE (PubMed), Scopus, and Web of Science. Supplementary approaches, including checking the reference lists of included or relevant sources of evidence, and searching trial registries or regulatory websites, were advocated. Any missing or unpublished information was acquired by contacting authors or sponsors. The search strategies were drafted and further refined through team discussion. Duplicates were removed by exporting the final search results into EndNote.

### 2.3. Search

The following search terms were used to search the databases to locate the articles needed for this review: ((“Vitamin D”[Mesh] OR “Vitamin D”[tw]) OR (“25-Hydroxyvitamin D 2”[Mesh] OR “Calcifediol”[Mesh] OR “Cholecalciferol”[Mesh] OR “Ergocalciferols”[Mesh] OR “25-hydroxyvitamin D”[tw] OR “25(OH)D”[tw] OR ”25-hydroxyvitamin D2”tw] OR “25-OH-D2”[tw] OR “25(OH)D2”[tw] OR “25D2”[tw] OR ergocalciferol[tw] OR “25-hydroxyvitamin D3”[tw] OR “25(OH)D(3”[tw] OR “25(OH)VD3”[tw] OR “25OHD”[tw] OR “25-OHD”[tw] OR “25(OH)D”[tw] OR “25D3”[tw] OR cholecalciferol[tw] OR calcifediol[tw]) OR (“1,25-dihydroxyvitamin D” [Supplementary Concept] OR “1,25-dihydroxyergocalciferol” [Supplementary Concept]) OR “Calcitriol”[Mesh] OR 1,25-dihydroxyvitamin D[tw] OR 1,25(OH)2D[tw] OR “1,25(OH)(2) D”[tw] OR 1,25-(OH)2)-D2[tw] OR “1alpha,25-Dihydroxyvitamin D”[tw] OR 1alpha,25(OH)(2)D[tw] OR “1,25-dihydroxyvitamin D2”[tw] OR “1alpha,25-dihydroxy vitamin D2”[tw] OR 1alpha,25(OH)2D2[tw] OR 1,25-dihydroergocalciferol[tw] OR “1,25-dihydroxyvitamin D3”[tw] OR 1,25(OH)(2)D(3)[tw] OR “1alpha,25-dihydroxy vitamin D3”[tw] OR 1alpha,25(OH)2D3[tw] OR calcitriol[tw])) AND (“Vitamin D Deficiency”[Mesh] OR “Hypovitaminosis D”[tw]) AND (“hypervitaminosis D” OR “vitamin D excess*”[tw] OR “vitamin D toxicity”[tw]) AND (“acute kidney injury”[MeSH Terms] OR acute kidney injury[tw] OR AKI[tw] OR ARF[tw])

The final search strategy for PubMed is as follows:

PubMed search strategy (literature search performed: 1 June 2024) 
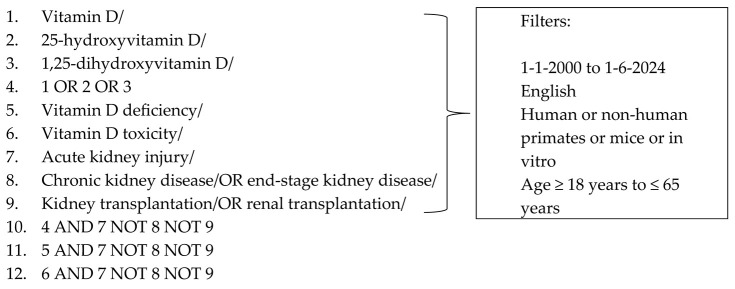


### 2.4. Selection of Sources of Evidence

A single reviewer conducted the eligibility assessment in an unblinded, standardized manner. The screening and data extraction manuals were discussed with two supervisors before screening for this review. The titles, abstracts, and entire texts of all publications searched were used to screen the records.

### 2.5. Data Charting/Extraction

To extract data from eligible studies, a standardized electronic data abstraction tool (based on “collection data” as outlined in Cochrane Training) [8] was employed. A single reviewer independently charted relevant information to determine the mutual effects of hypovitaminosis D and hypervitaminosis D in AKI and discussed and updated the supervisors in an iterative process.

### 2.6. Synthesis of Data

The studies were categorized based on the normal vitamin D homeostasis in humans and the impact of vitamin D deficiency and excess states on AKI and vice versa. When a systematic review was found, the number of studies included in this review that met the inclusion criteria, as well as those that were overlooked by the literature search, were counted. The evidence was given in the form of a narrative.

## 3. Role of Vitamin D in Health

Vitamin D is crucial for life in higher-living organisms. The association of vitamin D with rickets was discovered in 1924 [9]. Since then, it has been intensely researched with regard to its biological activities and its role in many diseases. In technical terms, vitamin D is a fat-soluble secosteroid because one of the rings of its cyclopentanoperhydrophenanthrene structure consists of a broken 9,10 carbon–carbon bond (ring B). Ergocalciferol (vitamin D2) and cholecalciferol (vitamin D3) differ only in their side chain structure, which determines their binding ability to vitamin D-binding protein (DBP), as well as their potency (Figure 1). Vitamin D3 is relatively more efficacious. In a study involving 20 healthy human subjects, Armas et al. (2004), showed vitamin D2 to be one-third less efficacious in relation to vitamin D3 [10]. Whereas vitamin D3 is derived exogenously from a diet consisting of animal-based foods such as fish oils, and endogenously from de novo epidermal synthesis, vitamin D2 is obtained mostly from plant sources and fortified food [1].

### 3.1. Vitamin D Synthesis, Transport, and Bioavailability

UV rays (270–300 nm) photolytically convert 7-dehydrocholesterol in the epidermis by breaking its B ring to form pre-vitamin D3, which then undergoes thermal isomerization to vitamin D3, or to tachysterol and lumisterol, with continued ultraviolet irradiation [11]. The cutaneous production of vitamin D3 varies seasonally and geographically and is also influenced by an individual’s social habits [12].

Vitamin D3 is preferentially removed from the skin by DBP and then hydroxylated in the liver by the cytochrome P450s (microsomal *CYP2R1* and mitochondrial *CYP27A1*) [2] to form 25-hydroxyvitamin D3, the principal circulating form of vitamin D. Approximately 90 percent of 25-hydroxyvitamin D3 circulates binding to DBP, around 10–15% binds to albumin, and less than 1% does not bind and circulates freely [13]. In comparison to 25-hydroxyvitamin D3, 1,25-dihydroxyvitamin D3 has a lower DBP-binding affinity. Similarly, 25-hydroxyvitamin D2 binds DBP with a lower affinity than 25-hydroxyvitamin D3 due to a structural difference at carbon position 24. Therefore, 25-hydroxyvitamin D2 is cleared rapidly from the circulation, leading to its reduced conversion to 1,25-dihydroxyvitamin D2. Consequently, the efficacy of vitamin D2 supplements to maintain the optimal serum 25-hydroxyvitamin D levels is reduced when compared with that of vitamin D3 [14].

DBP binding enables the inactive metabolite 25-hydroxyvitamin D3 to undergo glomerular filtration and subsequently be endocytosed by megalin-cubilin and the adapter protein disabled-2 (Dab2) in the apical membrane of the renal proximal tubular cells. Those cells lacking megalin and cubilin are engaged in the megalin-independent receptor-mediated uptake of DPB as well as through the non-receptor uptake of vitamin D, a process known as the free hormone hypothesis [13]. Free or bioavailable 25-hydroxyvitamin D3 levels depend upon the levels of DBP and other circulating vitamin D-binding serum proteins. Furthermore, the levels of DBP themselves vary according to gender and race. In the proximal tubules, 25-hydroxyvitamin D3 undergoes further hydroxylation by 1 α-hydroxylase (*CYP27B1*) to form the bioactive metabolite 1,25-dihydroxyvitamin D3 or calcitriol [3] (Figure 2).

### 3.2. One Alpha-Hydroxylase

1 α-hydroxylase, which is a constituent of the cytochrome P450 system (*CYP27B1*), is the rate-limiting enzyme functioning as a mixed-function oxidase. Its expression is predominant in the proximal renal tubular cells and is primarily found in the inner mitochondrial membrane. In addition, it is also active in the distal nephron and extrarenal tissues such as monocytes, keratinocytes, the colon, and lung and parathyroid cells. The first extrarenal 25-hydroxyvitamin D3-1 α-hydroxylase activity was demonstrated by Barbour and colleagues in 1981 in an anephric patient with sarcoidosis and hypercalcemia [15].

Of note, the renal 1 α-hydroxylase exerts endocrine actions while the extrarenal 1 α-hydroxylase largely functions in an autocrine/paracrine manner, with cell-specific activities [16]. The mRNA expression of 1 α-hydroxylase is determined by the plasma concentrations of calcium, phosphorus, 1,25-dihydroxyvitamin D3, parathormone (PTH), calcitonin, and bone-derived fibroblast growth factor 23 (FGF23) [17].

### 3.3. Mechanism of Vitamin D Action

Calcitriol acts via the vitamin D receptor (VDR), a nuclear receptor that functions as a transcription factor. Following calcitriol binding to the VDR, a heterodimer is formed with the retinoid X receptor (RXR) that binds to the vitamin D response element (VDRE) in the promoter gene region, resulting in an altered chromatin structure and induction of coactivators as well as co-repressors to modulate target gene expression [18]. The first zinc finger region of the VDR is crucial for homodimerization, while regions beyond this domain are essential for heterodimerization with RXRs. The carboxy-terminal region also facilitates heterodimer formation, and ligand binding enhances dimerization activity in this domain [19] (Figure 3).

In addition to its genomic effects, calcitriol also exerts non-genomic effects mediated by membrane-initiated signaling pathways, influencing second messengers and downstream processes [20,21] (Figure 4). These mechanisms account for its broad physiological actions, as *VDR* expression is nearly ubiquitous across tissues. Chromatin immunoprecipitation sequencing (ChiPseq) studies reveal 1000 to 10,000 VDR-binding sites in the human genome [22], underscoring its role in immune modulation and tissue-specific functions [23].

**Figure 3 biomolecules-15-00586-f003:**
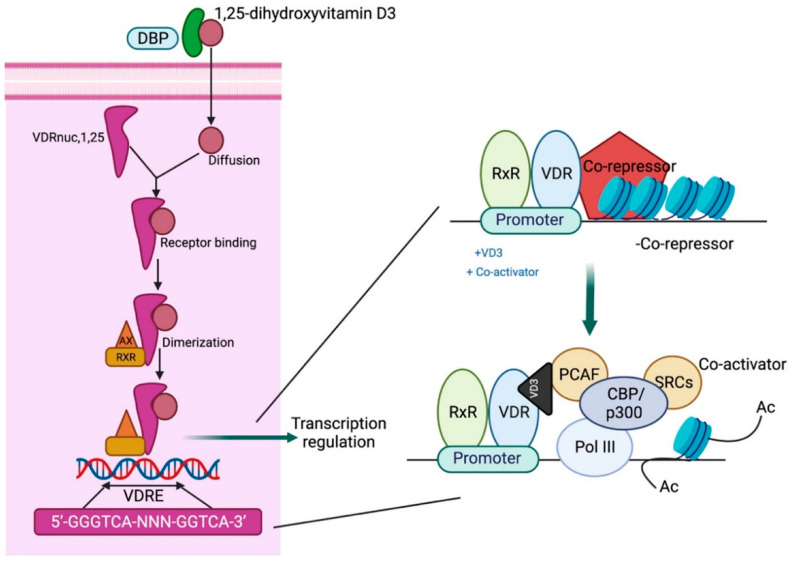
Mechanistic pathway of vitamin D action. DBP: vitamin D-binding protein; VDRE: vitamin D response element; CBP/p300: CREB-binding protein-binding protein p300; PCAF: P300/CBP-associated factor; SRC: steroid receptor coactivators. Adapted from Gil A et al. [21].

**Figure 4 biomolecules-15-00586-f004:**
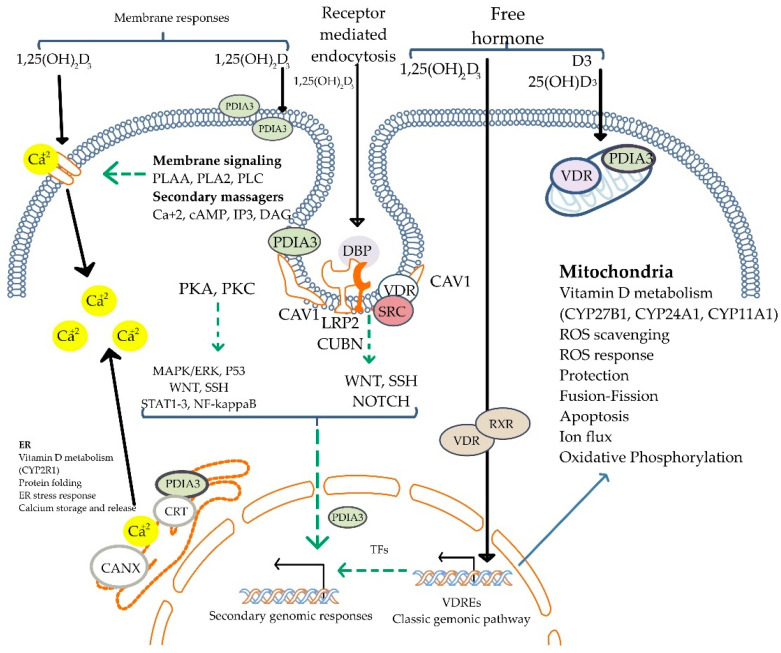
Mechanisms of non-genomic actions of Vitamin D. The active form, 1,25(OH)_2_D_3_, initiates rapid cellular responses by binding to membrane-associated vitamin D receptors (PDIA3 and VDR). These interactions activate intracellular signaling pathways, including PKA, PKC, MAPK/ERK, STAT3, and others, often via second messengers such as calcium ions (Ca^2^⁺), cAMP, IP_3_, and DAG. Vitamin D signaling also modulates WNT, NOTCH, and NF-κB pathways. Additionally, vitamin D influences mitochondrial functions and endoplasmic reticulum stress responses. The figure shows both receptor-mediated endocytosis and free hormone entry, reflecting the dual mechanism of membrane signaling and classic genomic pathway activation. 1,25(OH)_2_D_3_: 1,25-dihydroxyvitamin D_3_; VDR: vitamin D receptor; PDIA3: protein disulfide isomerase A3; PKA: protein kinase A; PKC: protein kinase C; MAPK/ERK: mitogen-activated protein kinase/extracellular signal-regulated kinase; STAT3: signal transducer and activator of transcription 3; NF-κB: nuclear factor kappa B; cAMP: cyclic adenosine monophosphate; IP_3_: inositol 1,4,5-triphosphate; DAG: diacylglycerol; Ca^2^⁺: calcium ion; WNT: wingless/integrated pathway; NOTCH: Notch signaling pathway; SRC: proto-oncogene tyrosine-protein kinase Src; DBP: vitamin D-binding protein; CUBN: cubilin; LRP2: low-density lipoprotein receptor-related protein 2 (megalin); CAV1: caveolin-1; RXR: retinoid X receptor; VDREs: vitamin D response elements; TFs: transcription factors; ROS: reactive oxygen species; CRT: calreticulin; CANX: calnexin; ER: endoplasmic reticulum. Adapted from Żmijewski MA et al. [24].

The dimerization of vitamin D receptors (VDRs) exhibits distinct patterns in health and kidney injury. In the context of kidney health, VDR activation plays a protective role through mechanisms such as suppression of the renin-angiotensin system (RAS) activation, anti-inflammatory effects, inhibition of renal fibrogenesis, mitochondrial function restoration, suppression of autoimmunity, and prevention of renal cell apoptosis [19]. These are influenced by multiple signal transduction pathways mediated through the genomic and non-genomic effects of the VDR in a cell-specific fashion [20,25]. These beneficial effects of VDR activation have been demonstrated in conditions such as IgA nephropathy, diabetic nephropathy, and lupus nephritis [26].

Conversely, dysfunctional VDR processes lead to resistance to vitamin D and other hormones [27]. Uremic toxins are shown to disrupt VDR synthesis, binding, and function, contributing to chronic kidney disease pathogenesis. Therefore, understanding the molecular dynamics of VDR dimerization in health and disease may guide the development of novel interventions for renal disorders [28]. For instance, vitamin D treatment has shown efficacy in reducing inflammation and myofibroblast formation in kidney ischemia/reperfusion injury [29]. This intricate interplay of genomic and non-genomic actions of calcitriol, modulated by VDR activity, underscores its multifaceted role in maintaining health and managing disease.

### 3.4. Biological Effects

Calcitriol exerts a cardinal function of maintaining calcium and phosphorus homeostasis by modulating parathormone secretion and increasing the gut absorption of calcium. Apart from this, it has extra-endocrine pleiotropic actions on various cellular functions, including proliferative, differentiating, apoptotic, cytoprotective, and reparative processes [5,30,31].

Distinctly, calcitriol downregulates adaptive and upregulates innate immunity, thereby it has a role in immunomodulation and reducing inflammation [32]. As aforementioned, it also possesses antioxidant, anti-inflammatory, antifibrotic, and antineoplastic properties and inhibits the renin-angiotensin-aldosterone system (RAAS) and NF-kB (nuclear factor kappa-light-chain-enhancer of activated B cells) activities [31,33].

### 3.5. Vitamin D Regulation

Calcitriol regulation is determined by the balance between 1 α-hydroxylase (*CYP27B1*) and vitamin D-24 hydroxylase (*CYP24A1*) activity. Calcium and phosphorus levels, as well as the interplay of positive and negative feedback loops involving PTH and fibroblast growth factor (FGF) and calcitriol itself, tightly control these enzymes [34]. In conditions of reduced serum calcium levels, PTH is synthesized and released by the parathyroid gland, which stimulates renal *CYP27B1* via cyclic adenosine 3′5′ monophosphate (cAMP) and also degrades *CYP24A1* mRNA in the kidney [35]. These result in enhanced calcium absorption from the gastrointestinal tract, increased calcium reabsorption in the kidney, and the stimulation of bone calcium release, thereby restoring serum calcium levels. Subsequently, the PTH is inhibited by calcitriol and FGF23 by a feedback mechanism.

FGF23 is synthesized and secreted by the osteocytes and is positively regulated by calcitriol and serum phosphorus concentrations. FGF23 further causes the inhibition of *CYP27B1* and the decreased renal expression of sodium-phosphate transporters in the presence of raised phosphate levels [36]. Finally, 1,25-dihydroxyvitamin D3 can inhibit its own synthesis by inhibiting PTH, repressing transcription of the *CYP27B1* mRNA, and inducing *FGF23* and *CYP24A1*. *CYP27B1* is also stimulated by insulin-like growth factor type 1 (IGF-1) and calcitonin [37,38].

### 3.6. Inactivation of Vitamin D

The catabolism of 25-hydroxyvitamin D3 and calcitriol into less metabolically active metabolites occurs principally through alterations in the expression of cytochrome enzymes. Vitamin D-24 hydroxylase (*CYP24A1*) catalyzes a five-step metabolic pathway, generating a water-soluble end product: calcitroic acid. Another 23-hydroxylation pathway, which is also catalyzed by *CYP24A1*, exists, resulting in the formation of the 1α,25-dihydroxyvitamin D3-26-23-lactone end product [39]. Calcitroic acid is conjugated in the kidney by glucuronidation, sulfation, methylation, and with amino acid and glutathione followed by excretion [40]. While renal excretion is less than 5%, the predominant excretion occurs via bile. These processes occur in a negative feedback mechanism to limit vitamin D toxicity. For instance, *CYP24A1* gene mutations have been implicated in idiopathic infantile hypercalcemia with inappropriately elevated calcitriol levels due to its impaired catabolism [41].

### 3.7. Vitamin D Analogs

Synthetic vitamin D analogs are therapeutically used for hormone replacement. Originally 1α-hydroxyvitamin D3 and 1 α-hydroxyvitamin D2 requiring just the 25-hydroxylation to render them bioactive were synthesized. Calcitriol occupies only 56% of the VDR ligand binding site. Therefore, analogs can be structurally modified to enable them to bind to the VDR. Notably, the A-ring, 149 and 150 C/D rings, and the side chain of 1,25-dihydroxyvitamin D3 are modified to form bioactive analogs [42].

### 3.8. Analysis and Quantification of Vitamin D Metabolites

Several methods are utilized in determining vitamin D concentrations in biological samples. Competitive protein binding assay, radioimmunoassay, ELISA, high-performance liquid chromatography (HPLC), and liquid chromatography-tandem mass spectrometry (LC-MS/MS) are some of the important analytical techniques. 25-hydroxyvitamin D is present abundantly in the plasma. Therefore, it is commonly used in assessing the body’s vitamin D status [43]. Because of high specificity and sensitivity, LC-MS/MS analysis is regarded as the ‘gold standard’ method and is used in the high throughput analysis of 25-hydroxyvitamin D3 as well as other vitamin D metabolites. Matrix-assisted laser desorption/ionization-mass spectrometry imaging (MALDI-MSI) is an analytical method used to map metabolites on tissue surfaces [44].

## 4. Role of Vitamin D in AKI

A two-way relationship between vitamin D and acute kidney injury (AKI) appears to exist [45]. Acute renal injury can cause hypo- or hypervitaminosis D in a variety of ways, including disruptions in hydroxylation pathways and altered hormonal regulation [46,47,48,49]. Contrarily, AKI can be caused by both vitamin D deficiency as well as vitamin D toxicity. Severe deficiency has been associated with the increased incidence and poor prognosis of AKI in critically ill patients [50], whereas excessive dosing of vitamin D has been shown to precipitate AKI in clinical case series [51,52]. Because our understanding of the association between vitamin D and AKI is based on limited case reports, series, and experimental models, drawing inferences about the broader population could be detrimental and misleading [29,50,53]. Furthermore, using vitamin D to treat AKI is problematic from a therapeutic standpoint because not many prospective randomized trials are available, and our understanding is based on animal investigations [53,54,55].

### 4.1. Dysregulation of Vitamin D in AKI

Biologically active vitamin D_3_, the calcitriol, is synthesized in the kidneys [3,56]. A complex relationship is also observed between vitamin D, AKI, and adverse outcomes [29,50]. AKI is characterized by a substantial loss of renal function, which interferes with normal renal enzymatic activity and, as a result, affects vitamin D metabolism [45]. In most cases, decreased kidney function results in vitamin D deficiency [46,47,48]; however, in some instances, vitamin D toxicity is also reported [49].

#### 4.1.1. Development of Hypovitaminosis D in AKI

Vitamin D deficiency occurs commonly in chronic kidney disease and is associated with secondary hyperparathyroidism, low hemoglobin levels, erythropoietin-resistant state, altered immunity, and enhanced risk of mortality and morbidity. In the AKI setting, vitamin D deficiency occurs due to several reasons. AKI causes phosphate retention due to renal function loss. Phosphate negatively regulates 1-hydroxylase, an enzyme that synthesizes 1,25(OH)2D, thereby affecting active vitamin D production and intestinal calcium absorption [46].

Lower calcitriol levels in patients with AKI have been attributed to reduced 25-hydroxyvitamin D levels, as well as renal dysfunction due to which less 25-dihydroxyvitamin D substrate is converted to active 1,25-dihydroxyvitamin D. Moreover, increased FGF23 levels could cause reduced activation of 25-hydroxyvitamin D by inhibiting 1 α-hydroxylase and induction of the catabolic 24-hydroxylase, leading to the impaired synthesis of calcitriol from 25-hydroxyvitamin D3 [47]. In addition, deficiency of klotho—more commonly observed in chronic kidney disease (CKD) but also present in AKI—contributes to vitamin D insufficiency through two mechanisms: (1) by exacerbating AKI due to the loss of its inherent renoprotective functions, and (2) by promoting FGF-23–mediated suppression of vitamin D activation [48].

#### 4.1.2. Development of Hypervitaminosis D in AKI

The depletion of calcium concentration causes an elevation in PTH levels. Consequently, the activation of *CYP27B1* and secondary hyperparathyroidism can both increase Vitamin D levels (hypervitaminosis D). An increase in 25(OH)D and 1,25(OH)2D has also been reported during the diuretic phase of AKI due to rhabdomyolysis, especially in those who developed hypercalcemia [49].

### 4.2. Vitamin D Deficiency and AKI

#### 4.2.1. Vitamin D and Critical Illness-Related AKI

Vitamin D deficiency is highly prevalent in critically ill patients, according to several studies. In a study by Zapatero et al. (2018), roughly 74% of 135 ICU patients had low 25-hydroxyvitamin D concentrations, which was associated with a 2.86-fold increase in the incidence of AKI and mortality [50].

Similarly, Lai et al. (2013) [57] compared 200 patients with AKI to healthy individuals and critically ill patients without AKI. They found that nearly all patients with AKI exhibited significantly reduced serum levels of 1,25-dihydroxyvitamin D (calcitriol), with a mean of 59.6 ± 53.0 pmol/L, compared to 86.2 ± 35.3 pmol/L in critically ill controls and 98.8 ± 39.7 pmol/L in healthy subjects (ANOVA, *p* = 0.005) [57]. Importantly, the severity of AKI correlated inversely with calcitriol levels. The study reported a progressive decline in calcitriol concentrations across RIFLE stages, with mean values of 72.6 ± 69.4 pmol/L in the Risk group, 53.7 ± 32.7 pmol/L in the Injury group, and 42.2 ± 29.3 pmol/L in the Failure group (*p* = 0.042). This represents a relative decline of approximately 42% between the Risk and Failure groups, indicating a severity-dependent reduction in calcitriol levels. Notably, 25-hydroxyvitamin D levels did not differ significantly between groups, and vitamin D status—when adjusted for age, gender, SOFA score, and VDR polymorphisms—did not predict 90-day mortality in multivariate Cox regression analysis [57].

Importantly, despite various RCTs (randomized controlled trials) reporting the normalization of serum vitamin D concentrations following vitamin D supplementation [58], conflicting outcomes have been observed in relation to patient survival and hospitalization rates in the critical care setting [54,59]. Several meta-analyses have also reported discordant findings, thereby precluding routine vitamin D supplementation in seriously ill patients with deficiency [53,55]. Vijayan A et al. (2015) similarly demonstrated a positive correlation between raised calcitriol concentrations and mortality and the need for renal replacement therapy. They conjectured that bioactive vitamin D with its antiproliferative and pro-differentiating effects could prolong AKI by delaying tissue recovery [60]. In this context, it is worth considering that the regeneration and recovery of kidneys from ATN (acute tubular necrosis) entails de-differentiation and proliferation of renal tubular epithelial cells. Furthermore, macrophages are activated in sepsis and stimulate extrarenal production of 1,25-dihydroxyvitamin D. Consequently, excess 1,25-dihydroxyvitamin D levels can theoretically be associated with increased mortality [61]. Based on these contradictory observations, it is unclear whether vitamin D deficiency causes AKI and increased mortality or if it is just an indicator of disease severity, and is a matter of ongoing debate [50].

#### 4.2.2. Vitamin D and Sepsis-Induced AKI

AKI is one of the important complications of sepsis. Lipopolysaccharide (LPS) is an endotoxin derived from a gram-negative bacterial wall and is a potent inducer of sepsis by activating NF-kB through its interaction with toll-like receptor subtype 4 (TLR4), a key innate immune receptor. NF-kB is crucial for regulating renal inflammation [62].

Previously, researchers have shown the suppression of lipopolysaccharide-induced pro-inflammatory cytokines in the renal tubular epithelial cells by vitamin D [63]. Further, in studies on mice and HK2 cells, Du et al. (2019) [64] demonstrated that vitamin D blocks renal tubular epithelial inflammation and apoptosis induced by lipopolysaccharide/toll-like receptor 4 (LPS/TLR4) pathway. Mice without VDR suffered intense renal injury and an increase in renal cellular apoptosis compared to wild-type control mice following exposure to lipopolysaccharide. Additionally, Bcl-2 was observed to be downregulated, p53-upregulated modulator of apoptosis (PUMA) was vigorously induced, and caspase-3 was drastically activated in the renal cortex of the VDR-knockout mice. All these were abrogated by paricalcitol treatment. Similarly, in HK2 cells, lipopolysaccharide was found to induce PUMA and miR-155 by NF-kB, which were inhibited by 1,25-dihydroxyvitamin D3. Of note, apoptosis is promoted by both PUMA and miR-155 by inhibiting Bcl-2 activity and Bcl-2 protein translation, respectively [64].

#### 4.2.3. Vitamin D and Contrast-Induced AKI

Paricalcitol is a bioactive, non-hypercalcemic vitamin D analog with efficacy equivalent to that of vitamin D and relatively lesser untoward effects. It is chemically 19-nor-1,25-dihydroxyvitamin D2. Apart from possessing antioxidant properties, it is known to suppress the RAAS in the kidneys [65]. Ari et al. (2012) [66] discovered that giving paricalcitol 4 days before using a contrast agent protected Wistar albino rats from developing contrast-induced renal damage, as evidenced by decreased serum creatinine levels and a rise in creatinine clearance. Paricalcitol was also shown to circumvent oxidative stress by markedly reducing MDA (Malondialdehyde) and thiobarbituric acid reactive substances (TBARS) levels. Histologically, the mean scores of tubular necrosis, protein casts, congestion of the renal medulla, and vascular endothelial factor (VEGF) expression were remarkably lower [66].

Vitamin D deficiency is known to be associated with increased RAAS activity, oxidative stress, and endothelial dysfunction. Interestingly, Luchi et al. [67] observed that healthy rats treated with contrast media exhibited no altered redox potential, maintained a normal glomerular filtration rate (GFR), and showed enhanced endothelial nitric oxide synthase (eNOS) levels. These findings suggest that an additional risk factor, such as vitamin D deficiency, is necessary to induce contrast-induced acute kidney injury (CI-AKI). Furthermore, vitamin D-deficient rats experienced greater oxidative stress, as evidenced by elevated renal parenchymal and urinary thiobarbituric acid reactive substances (TBARS), along with reduced renal and systemic glutathione (GSH) levels [67].

The histological alterations in the kidneys caused by CI-AKI have been variable among investigations. The majority of the researchers found no significant morphological alterations in the kidneys, while some found proximal tubular vacuolization with no link to kidney disease [68]. Less severe tubular injury without macrophage infiltration was highlighted by Luchi et al. (2015), indicating that the intrarenal hemodynamic alterations led to a decrease in inulin clearance primarily in the kidneys in the absence of inflammation [67]. Worthwhile, these authors also showed that the influence of contrast media on renal functions in vitamin D deficiency was independent of the osmotic load and the inulin clearance was inversely related to the duration of vitamin D deficiency state.

#### 4.2.4. Vitamin D and Aminoglycoside-Induced AKI

Medication-related nephrotoxicity is an alarming and growing problem, accounting for almost 10–20 percent of acute kidney injury cases, with 2–7 percent of in-patients being affected [69]. It is vital to remember that 25 percent of the 100 most regularly used medications in ICUs are potentially nephrotoxic [70]. Despite preventive measures such as sufficient hydration and observation, the aminoglycoside gentamicin causes kidney injury in 10–25 percent of cases. Renotoxicity involves injury to multiple renal compartments, including the tubules, glomeruli, and renal vasculature, and is known to be dose-dependent [71].

Another commonly used aminoglycoside antibiotic, amikacin, accumulates in the proximal tubule and causes nephrotoxicity by generating free radicals and oxidative stress, increased endothelin-1 and transforming growth factor beta (TFG-β) levels, monocyte/macrophage infiltration of the renal parenchyma, tubular epithelial brush border injury, and tissue necrosis [71]. In addition, gentamycin increased caspase-3 and Jun-N-terminal kinase activity in rat kidneys, which were successfully reduced by paricalcitol [72]. Further, Hur et al. (2013) found that vitamin D can protect against gentamicin nephrotoxicity by increasing glutathione levels [73].

#### 4.2.5. Vitamin D and Cisplatin-Induced AKI

Cisplatin, an antineoplastic drug can cause nephrotoxicity and has been found to result in substantial elevations of bleomycin-detectable iron in the renal tissue that could be ameliorated by the administration of an iron chelator such as deferoxamine [74].

In a cisplatin-induced AKI model, Hu et al. (2020) [75] employed ferrostatin-1 to prevent ferroptosis and found a reduction in BUN and serum creatinine. Paricalcitol, a VDR agonist, lowered malondialdehyde and 4-hydroxynonenal (4-HNE) levels while maintaining glutathione peroxidase 4 (GPX4) activity, a prime regulator of ferroptosis, thereby ameliorating cisplatin nephrotoxicity. VDR-knockout mice, on the other hand, showed severe ferroptosis and kidney damage when compared to wild-type mice. Furthermore, in both in vitro and in vivo cisplatin-induced AKI models, VDR downregulation significantly reduced *GPX4* expression. Importantly, *GPX4* was discovered to be the transcription factor VDR’s target gene [75]. Furthermore, small interfering RNA (siRNA) was shown to inhibit *GPX4* expression, thereby eliminating the protective effect of paricalcitol in cisplatin-induced renal injury. Outside of this, pre-treatment with paricalcitol prevented Erastin-induced ferroptosis in HK-2 cells. These data suggest that by overcoming ferroptosis, VDR activation may be able to prevent cisplatin-induced kidney injury [75].

#### 4.2.6. Vitamin D and Rhabdomyolysis

Rhabdomyolysis is a clinical syndrome characterized by the breakdown of skeletal muscle tissue, leading to the release of intracellular contents such as myoglobin into the bloodstream, which can contribute to renal injury. AKI complicates around 10–40 percent of cases with rhabdomyolysis, with a mortality rate of nearly 59 percent [76]. The kidney involvement in rhabdomyolysis is better understood by employing a glycerol-induced AKI animal model [77].

The glycerol-mediated rhabdomyolysis rat model is characterized by skeletal muscle breakdown and the release of intracellular components such as myoglobin into the bloodstream, where it contributes to renal tubular injury through filtration, reabsorption, and distal precipitation [78]. In this model, experimental rats exhibited elevated serum creatine kinase levels, augmented fractional sodium excretion, increased urine output, decreased glomerular filtration rate (GFR), and reduced urine osmolality. Calcitriol administration reversed these findings in addition to decreasing the levels of isoprostane, an oxidative stress marker, and nitrotyrosine, a protein nitration marker, and increasing the antioxidant superoxide dismutase activity. Furthermore, these calcitriol-administered rats were found to preserve cubilin receptors, emphasizing the nephroprotection offered by it [78].

#### 4.2.7. Vitamin D and Ischemia-Reperfusion Injury

Ischemia-reperfusion injury (IRI) disrupts renal tubular cells and causes AKI. It can further lead to fibrosis and eventually culminate in chronic kidney disease in 70 percent of cases [79]. A complex interplay of renal hemodynamic changes, tubulotoxicity, inflammation, cell proliferation, oxidative, endoplasmic and mitochondrial stress, and apoptosis plays a role in its pathogenesis [80]. Furthermore, a deficiency of vitamin D (VDD) increases nitric oxide synthesis, reduces macrophage infiltration, suppresses adhesion molecular expression in the endothelium, and causes endothelial dysfunction [81]. De Braganca et al. (2015) also demonstrated VDD to potentiate IRI-induced AKI in rats, and progression to CKD by promoting inflammation and fibrosis [82].

Mice pre-treated with cholecalciferol appeared to alleviate IRI by inhibiting renal tubular cell apoptosis, endoplasmic and oxidative stress, inflammation, and fibrosis. Vitamin D inhibits renal fibrosis through multiple mechanisms, including direct interaction with Smad3 to block TGF-β–Smad signal transduction, and by independently stimulating the expression of hepatocyte growth factor *(HGF*) in the liver, which prevents renal myofibroblast generation. Additionally, vitamin D suppresses the *TGF-β1*-induced expression of fibrotic markers such as α-SMA, type I collagen, and thrombospondin-1 [83].

Moreover, vitamin D has been demonstrated to be beneficial in the prevention of active Heymann nephritis [84] and lupus in experimental mice [85], the reduction of podocyte loss [86] and glomerulosclerosis [87] in rats following subtotal nephrectomy, lowering albuminuria in experimental diabetic nephropathy [88], and attenuating interstitial fibrosis in obstructive uropathy [89]. Furthermore, it retards renal failure progression in uremic rats [90].

### 4.3. Hypervitaminosis and AKI

Vitamin D level assays have become some of the most routinely ordered laboratory procedures in the last two decades. This rise is attributable to an increasing awareness of widespread vitamin D insufficiency, as well as scientific evidence pointing to beneficial effects of vitamin D that extend beyond bone [91]. As the use of vitamin D therapy has grown, so has the number of cases of vitamin D intoxication, with the majority (75%) of reports published since 2010 indicating vitamin D intoxication [92]. Both pre-vitamin D3 and vitamin D3 are photolyzed to various non-calcemic photoproducts; hence, excessive exposure to sunlight will not result in vitamin D toxicity [93].

In one of the largest case series of AKI secondary to vitamin D intoxication, Wani et al. (2016) [51] reported 62 cases from the Indian Kashmir valley—a region where vitamin D deficiency is endemic and high-dose injectable vitamin D (600,000 IU per injection) is commonly overused. All cases involved patients with hypercalcemia and acute kidney injury (AKI) attributed to vitamin D toxicity. The cohort was divided into two groups: 51 patients with de novo AKI (group 1), and 11 patients with AKI superimposed on pre-existing chronic kidney disease (CKD) (group 2) [51].

Group 1 received between 4 and 28 injections of 600,000 IU, corresponding to a cumulative dose of 2.4 to 16.8 million IU of vitamin D. Group 2 received 3 to 24 injections (1.8 to 14.4 million IU). The mean serum creatinine at presentation was 3.2 ± 0.9 mg/dL in group 1 and 4.5 ± 1.1 mg/dL in group 2, indicating impaired renal function in both groups. Mean serum calcium levels were significantly elevated in both groups (13.7 ± 1.4 mg/dL in group 1 vs. 13.6 ± 2.0 mg/dL in group 2), findings that are atypical of uncomplicated AKI or CKD, but characteristic of vitamin D toxicity. Mean 25-hydroxyvitamin D levels were also markedly elevated (313.3 ± 54.8 nmol/L vs. 303.7 ± 48.4 nmol/L), strongly confirming the diagnosis of vitamin D toxicity [51]. Notably, parathyroid hormone (PTH) levels revealed distinct physiological responses in the two groups. In group 1 (de novo AKI), mean PTH was appropriately suppressed (18.1 ± 9.6 pg/mL), consistent with the expected feedback inhibition by hypercalcemia. In contrast, group 2 (AKI on CKD) showed elevated PTH levels (52.3 ± 12.6 pg/mL), a finding likely attributable to underlying secondary hyperparathyroidism due to CKD. In such patients, chronic reductions in calcitriol synthesis, phosphate retention, and reduced sensitivity of calcium-sensing receptors often lead to elevated baseline PTH levels that are not fully suppressed, even in the face of acute hypercalcemia. This pathophysiological mechanism explains the differential PTH response between the two groups. Common presenting symptoms included weakness, constipation, abdominal pain, nausea, vomiting, anorexia, altered sensorium, and oliguria. Management included intravenous saline in all group 1 patients and in most of group 2, short-term corticosteroids in 44 cases, and bisphosphonates in 6 patients. Most individuals showed clinical and biochemical improvement over a mean follow-up of 7.2 ± 0.6 months [51].

In yet another study, Sharma LK et al. discovered hypervitaminosis D (25-OHD > 250 nmol/L) in 225 (4.1%) patients, of whom 151 (2.7%) had vitamin D intoxication (25-OHD > 375 nmol/L). Orthopedic, pediatric, and surgical patients had the highest incidences of hypervitaminosis D (7.9, 7.2, and 7%, respectively; *p* < 0.001) [52]. In Brazil, 13 patients suffered AKI after receiving intramuscular injections of veterinary vitamins A, D, and E for aesthetic purposes [94].

Many of these occurrences are the result of excessive supplementation or over-fortified milk, improper prescribing, the use of high-dose over-the-counter or unlicensed medications, or even manufacturing defects [92,95]. Hypervitaminosis D can impair renal functions by causing hypercalcemia and hyperphosphatemia. Nephrogenic diabetes insipidus, which causes polyuria and diuresis, is induced by hypercalcemia, in particular. Thus, fluid loss produces hypovolemia, promotes AKI, and worsens hypercalcemia. Consequently, there is a vicious spiral in which hypercalcemia causes hypovolemia-induced AKI and vice versa. Hypercalcemia and hypercalciuria can further produce calcium deposition, nephrolithiasis, and renal calcification (nephrocalcinosis), which can lead to AKI. Hypercalcemia can also cause renal vasoconstriction, significant GFR reduction, and AKI [96]. As for hyperphosphatemia, it can develop acute phosphate nephropathy due to the tubulointerstitial deposition of phosphate and calcium, a situation that concomitantly worsens the already-existing hyperphosphatemia, thereby leading to a vicious circle of worsening. Acute phosphate nephropathy can also be brought on by a high phosphate intake and diarrhea-induced hypovolemia [97].

This literature review reveals that while vitamin D intoxication is uncommon, it does occur and is preventable; consequently, patients and prescribers should be more aware of the possible risks of vitamin D overdose, especially in individuals at risk of AKI [91]. Protracted, unsupervised, and impromptu vitamin D administration at non-recommended supraphysiological levels, particularly via intramuscular injections, should be avoided [52]. In this regard, Barth K. et al. reported two cases of hypercalcemia and acute kidney injury caused by vitamin D intoxication, successfully managed with denosumab. The rationale for its use lies in its mechanism of action as a monoclonal antibody against RANKL (Receptor Activator of Nuclear Factor κB Ligand). Excess active vitamin D increases RANKL expression by osteoblasts, which in turn stimulates osteoclast maturation and activity, leading to accelerated bone resorption and elevated serum calcium levels. Denosumab inhibits this RANKL–RANK interaction, thereby reducing osteoclast-mediated calcium release and counteracting the hypercalcemia induced by vitamin D toxicity [98].

To provide a comprehensive perspective, Table 1 summarizes the key clinical considerations regarding vitamin D dysregulation in AKI and its clinical implications.

Certain limitations in this study were unavoidable due to constraints in available data and methodological challenges. The lack of a direct discussion on vitamin D therapy approaches and recommendations for AKI stems from the heterogeneity of existing clinical trials and the absence of standardized guidelines. Similarly, reliance on experimental studies, primarily involving animal models and in vitro research, was necessary due to the limited availability of human data, restricting direct applicability to clinical practice. Additionally, the absence of quantitative data on vitamin D supplementation effect sizes and cut-off threshold values for vitamin D toxicity was influenced by the variability in reported outcomes.

Addressing these gaps in future studies through larger clinical trials, the integration of diverse patient cohorts, and the systematic evaluation of confounding factors such as comorbidities (e.g., diabetes, cardiovascular disease), concurrent medications, and ICU interventions on AKI outcomes and vitamin D metabolism will strengthen the understanding of vitamin D’s role in AKI management. Furthermore, a comparative analysis of vitamin D analogs, such as paricalcitol versus cholecalciferol, could provide insight into their differential effects in AKI settings. Moreover, personalized vitamin D therapy, guided by genetic markers and individual patient risk factors, may lead to more targeted and effective interventions for improving renal outcomes.

## 5. Conclusions

Vitamin D and acute kidney injury exhibit a complex, bidirectional (reciprocal) relationship. Hypo- or hypervitaminosis D may arise due to various factors, including acute renal injury. Conversely, AKI can be precipitated by both vitamin D deficiency and vitamin D intoxication. Calcitriol, the biologically active form of vitamin D3, is synthesized in the kidneys and plays a critical role in this dynamic. Furthermore, the interplay between vitamin D, AKI, and adverse clinical outcomes is complex and multifaceted.

Although calcitriol supplementation has demonstrated pleiotropic and renoprotective properties, its definitive role in preventing renal injury remains inconclusive. While hypovitaminosis D has been associated with the development of AKI, evidence regarding its impact on AKI outcomes across diverse clinical settings is limited. Addressing these gaps in understanding could pave the way for innovative therapeutic strategies.

On the other hand, hypervitaminosis D, often resulting from inappropriate prescribing practices or excessive use of over-the-counter or unregulated supplements, has been implicated in AKI. This condition commonly induces hypercalcemia and hyperphosphatemia, further exacerbating renal injury. Therefore, avoiding extended, unplanned, and uncontrolled supraphysiological vitamin D treatment—especially intramuscular administration—is imperative. Addressing these knowledge gaps may pave the way for novel therapeutic strategies in nephrology.

## Figures and Tables

**Figure 1 biomolecules-15-00586-f001:**
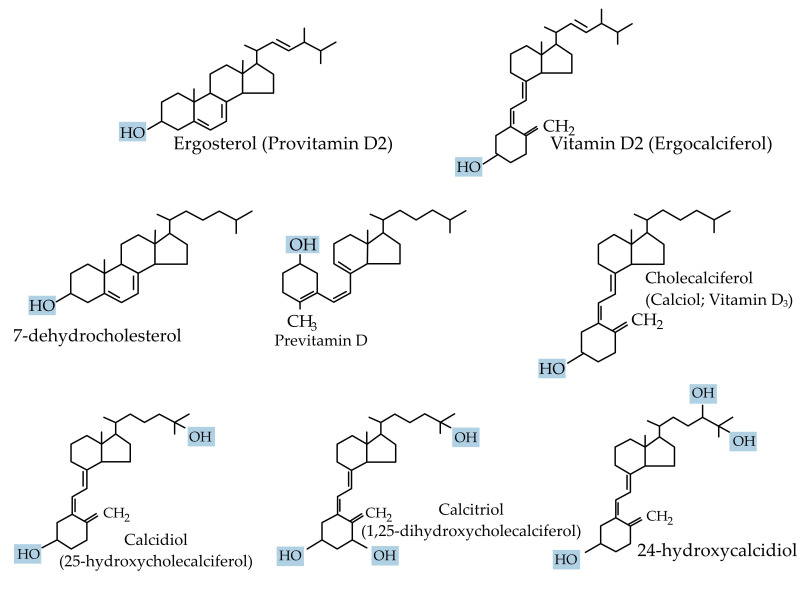
Chemical structure of the major forms of vitamin D.

**Figure 2 biomolecules-15-00586-f002:**
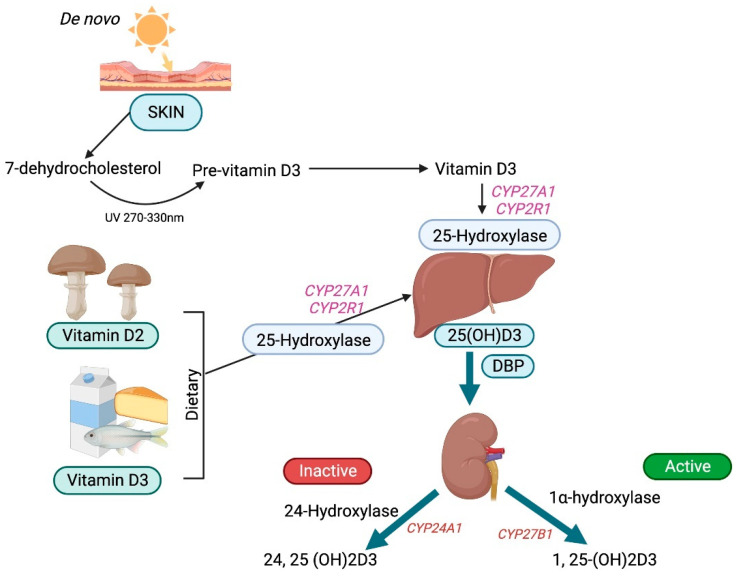
Schematic diagram of vitamin D metabolism.

**Table 1 biomolecules-15-00586-t001:** Clinical implications of vitamin D dysregulation in acute kidney injury (AKI).

Aspect	Current Evidence	Clinical Implications
Vitamin D Deficiency	Common in AKI, correlates with worse outcomes but lacks strong causal evidence [50,53,55,57,58,63,66,73,75,78,83]	Routine vitamin D monitoring in AKI patients may be beneficial but requires further study
Vitamin D Supplementation	Mixed results from RCTs, with some showing benefit and others no impact [53,54,55,58,60,61]	Individualized approach needed; avoid universal supplementation without monitoring
Hypervitaminosis D	Can cause hypercalcemia, nephrocalcinosis, and AKI, especially with high-dose IM injections [51,52,91,92,98]	Controlled administration essential; avoid excessive dosing

AKI, Acute kidney injury; RCTs, Randomized controlled trials, IM, Intramuscular.

## Data Availability

The original data presented in this study are openly available at the following URL: https://drive.google.com/drive/folders/1m7ZNNmLvri0Xiz5s30GVZKK0B9sjAFPh?usp=sharing (accessed on 6 April 2025).

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
