# Peer review of "Vitamin D and Acute Kidney Injury: A Reciprocal Relationship"

_biomolecules, 2025, doi:10.3390/biom15040586_

Round 1

Reviewer 1 Report

Comments and Suggestions for Authors

Manuscript entitled Vitamin D and Acute Kidney Injury: A Reciprocal Relationship describes nicely interaction that arise in AKI related to vitamin D levels. However, there are many minro points, some of which are listed and other need to be solves following suggestions.

Minor:

  • references are in square brackets AFTER dot. I am not sure it is correct way.
  • some references are mission or are not references, but numbers with no meaning (e.g. line 389)
  • there are some words that are certainly not meant to be there (e.g. line 371 "Few meta-analyses too have")
  • line 418 "oxidant stress"
  • rephrase line 441: Renotoxicity is caused by the interaction of tubules, glomeruli,...
    and the renal vasculature
Comments on the Quality of English Language

As above

Author Response

Comment 1: references are in square brackets AFTER dot. I am not sure it is correct way.

Response 1:

We thank the reviewer for pointing this out. As per the Instructions for Authors of the journal, reference numbers should indeed be placed before punctuation marks. We have carefully revised the manuscript and corrected the placement of all in-text citations to conform to this style (e.g., “[1],” “[1–3]”), ensuring that references now appear before commas and periods throughout the text.

Revised in-text citations in the manuscript

-        Updated throughout the manuscript

______________________________________________________________________________

Comment 2: some references are mission or are not references, but numbers with no meaning (e.g. line 389)

Response 2:

We thank the reviewer for pointing this out. Upon review, we believe the confusion may have arisen from the passage describing Toll-like receptor 4 (TLR4). In this case, the number “4” refers to the specific subtype of the receptor (TLR4) and is not intended to be a reference number. To avoid ambiguity, we have now revised the formatting in the manuscript to make it clearer that this is part of the scientific term and not a citation.

Original text (Page 10, para 1, line 389 in Manuscript Version 1)

4.2.2 Vitamin D and sepsis-induced AKI

AKI is one of the important complications of sepsis.  Lipopolysaccharide (LPS) is an endotoxin derived from gram-negative bacterial wall and is a potent inducer of sepsis by activating NF-kB through its interaction with toll-like receptor (TLR) 4. NF-kB is crucial for regulating renal inflammation.[57]

Revised text (Page 11, para 3, lines 462-463 in Manuscript Version 2 with track changes)

“...through its interaction with Toll-like receptor subtype 4 (TLR4), a key innate immune receptor…”

______________________________________________________________________________

Comment 3: there are some words that are certainly not meant to be there (e.g. line 371 "Few meta-analyses too have"

Response 3:

We thank the reviewer for this observation. The phrase in question, “Few meta-analyses too have”, was intended to convey that a small number of meta-analyses have also reported conflicting findings. However, we recognize that the original phrasing could be unclear or appear grammatically awkward. We have now revised the sentence for clarity and improved readability.

Original text (Page 9, para 4, lines 371-373 in Manuscript Version 1)

Importantly, despite various RCT’s (randomised controlled trials) reporting normalisation of serum vitamin D concentrations following vitamin D supplementation,[50] conflicting outcomes have been observed in relation to the patient survival and hospitalization rates in the critical care setting.[51,52]  Few meta-analyses too have provided discordant findings, thereby precluding vitamin D supplementation in seriously ill subjects deficient in vitamin D.[53,54]  Vijayan A et al. (2015) similarly demonstrated a positive correlation between raised calcitriol concentrations and mortality and the need for renal replacement therapy.  They conjectured that bioactive vitamin D with its antiproliferative and pro-differentiating effects could prolong AKI by delaying tissue recovery.[55]…

Revised Sentence (Page 11, para 2, lines 444-446 in Manuscript Version 2 with track changes)

“…Several meta-analyses have also reported discordant findings, thereby precluding routine vitamin D supplementation in seriously ill patients with deficiency…”

____________________________________________________________________________

Comment 4: line 418 "oxidant stress"

Response 4:

We thank the reviewer for drawing attention to this. The term “oxidant stress” was used as a variant of the more commonly accepted term “oxidative stress.” We agree that “oxidative stress” is the more standard and widely recognized terminology in the biomedical literature. Accordingly, we have revised the phrase to “oxidative stress” in the manuscript for clarity and consistency.

Original text (Page 10, para 4, lines 413 and 418-419 in Manuscript Version 1)

4.2.3 Vitamin D and contrast-induced AKI

Paricalcitol is a bioactive, non-hypercalcemic vitamin D analog with efficacy equivalent to that of vitamin D and relatively lesser untoward effects.  It is chemically 19-nor-1,25-dihydroxyvitamin D2.  Apart from possessing antioxidant property, it is known to suppress the RAAS in the kidneys.[59] Ari et al. (2012) discovered that giving paricalcitol 4 days before using a contrast agent protected Wistar albino rats from developing contrast-induced renal damage, as evidenced by decreased serum creatinine levels and a raise in the creatinine clearance.  Paricalcitol also was shown to circumvent oxidant stress by markedly reducing MDA and thiobarbituric acid reactive substances (TBARS) levels.  Histologically, the mean scores of tubular necrosis, protein casts, congestion of the renal medulla and vascular endothelial factor (VEGF) ex-pression were remarkably lower.[60]

Vitamin D deficiency is known to be associated with increased RAAS activity, oxidant stress and endothelial dysfunction. Interestingly, healthy rats treated with the contrast media had no altered redox potential but did have enhanced endothelial nitric oxide synthase (eNOS) levels as well as normal GFR. These data suggest that a second risk factor, such as vitamin D deficiency, is required for inducing CI-AKI…

Revised Sentence (Page 12, para 2 and line 488, para 3 and lines 495-496 in Manuscript Version 2 with track changes)

“…Paricalcitol also was shown to circumvent oxidative stress by…”

“Vitamin D deficiency is known to be associated with increased RAAS activity, oxidative stress, and endothelial dysfunction…”

__________________________________________________________________________

Comment 5: rephrase line 441: Renotoxicity is caused by the interaction of tubules, glomeruli,...and the renal vasculature

Response 5:

We thank the reviewer for this helpful suggestion. We agree that the original sentence was unclear and may have implied an incorrect mechanism. To better reflect the multifactorial nature of aminoglycoside-induced renal injury, we have rephrased the sentence for clarity and accuracy.

Original text (Page 11, para 1, 441-442 in Manuscript Version 1)

“…Renotoxicity is caused by the interaction of tubules, glomeruli, and the renal vasculature, and it is dose-dependent.[65]”

Revised Sentence (Page 12, para 1, lines 527-529 in Manuscript Version 2 with track changes)

“…Renotoxicity involves injury to multiple renal compartments, including the tubules, glomeruli, and renal vasculature, and is known to be dose dependent [71].”

________________________________________________________________________

Reviewer 2 Report

Comments and Suggestions for Authors

General Comments

The manuscript by Annamalai and Viswanathan reviews the role of Vitamin D in Acute Kidney Injury (AKI), and how vitamin D status influences the progression of AKI. Overall, the manuscript is well organized and written. It is a much-needed comprehensive review of this topic.  Impressively, the methods section carefully explains the procedures employed to identify the articles to be included in the review. There are 2 major sections in the review, including the role of vitamin D in health which facilitates our understanding of the second section, the role of vitamin D in AKI. A number of modifications are needed, as described below, prior to its publication.

Specific Comments

Lines 137-142.  A Figure showing the structures of the differ molecular forms of vitamin D is needed here.

Lines 209-211.  The signaling pathways for nongenomic effects are not shown in Figure 2. A separate Figure shown nongenomic effects should be included.

Lines 236-240. The authors state that vitamin D treatment has efficacy in reducing inflammation and myofibroblast formation in kidney ischemia/reperfusion injury. The authors should describe how an intricate interplay of genomic and non-genomic actions of calcitriol modulated by VDR activity plays a role in the above stated events.

Lines 311-18. The authors should site references in this section of the review.

Lines 321-326. The authors should also site references in this section of the review.

Lines 359-362.  What proportion of the individuals with AKI had a decrease in calcitriol levels, and what was the relative level of the decrease? In the next sentence (lines 362-363) the authors state that “these levels increased in tandem with the severity of AKI,” which is in contradiction to the previous sentence. To what extent do these levels decrease (or increase) with the severity of AKI?

Line 414 define “MDA”

Lines 419-421. A reference should be included at the end of the sentence.

Lines 464-465. A reference should be included at the end of the sentence.

Line 465. Wat the small interfering RNA (siRNA) referred to here against GPX4? Specify this in the manuscript.

Line 472. The authors should define what the syndrome, rhadomyolysis, is here.

Lines 476-477. Is myoglobin released into the blood in this experimental model system?

Line 500. Does TGF beta decrease the induction of alpha SMA biosynthesis by TGF beta1? Please clarify.

Line 524. Did group 2 receive 3 to 24 million units of vitamin D? Did the individuals in group 2 already have either acute or chronic AKI at the start of the study? Were the mean creatinine levels, serum calcium levels vitamin D levels and PTH levels typical of those observed in either acute or chronic renal disease?  The authors should comment on the elevated PTH levels observed in group 2, as opposed to group 1.

Line 562. What is the mode of action of denosumab?

Lines 567-569. References must be added to each section of Table 1

Author Response

Comment 1: Lines 137-142.  A Figure showing the structures of the differ molecular forms of vitamin D is needed here.

Response 1:

We thank the reviewer for this excellent suggestion. In response, we have now included a new figure (Figure 1) depicting the chemical structure of the key molecular forms of vitamin D, including precursors, intermediate metabolites, and the active hormone. This visual aid complements the descriptive text in lines 137–142 and improves the reader’s understanding of vitamin D metabolism.

Revised with the following figure (Page 3, para 5, line 142 in Manuscript Version 2 with tracking)

Figure 1. Chemical structures of the major molecular forms of vitamin D.

_______________________________________________________________________________

Comment 2: Lines 209-211.  The signaling pathways for nongenomic effects are not shown in Figure 2. A separate Figure shown nongenomic effects should be included.

Response 2:

We sincerely thank the reviewer for this constructive suggestion. We agree that the original Figure 2, adapted from Gil et al., predominantly depicted the genomic mechanisms of vitamin D action. To address this, we have now included a new figure (Figure 4) that specifically illustrates the non-genomic signaling pathways of vitamin D. This schematic highlights the membrane-bound VDR interactions, second messengers (e.g., calcium, cAMP, IP3, DAG), intracellular kinases (PKA, PKC, MAPK/ERK), and the involvement of signaling modules like WNT, STAT3, and NF-κB, which mediate rapid cellular responses. This addition complements the textual explanation in lines 226-229, page 6 of the revised manuscript and provides a clearer visual reference for the non-genomic mechanisms of action.

Revised and incorporatde the following figure (Page 7, para 1, line 235-254 in Manuscript Version 2 with tracking)

Figure 4. Mechanisms of non-genomic actions of vitamin D. The active form, 1,25(OH)₂D₃, initiates rapid cellular responses by binding to membrane-associated vitamin D receptors (PDIA3 and VDR). These interactions activate intracellular signaling pathways including PKA, PKC, MAPK/ERK, STAT3, and others, often via second messengers such as calcium ions (Ca²⁺), cAMP, IP₃, and DAG. Vitamin D signaling also modulates WNT, NOTCH, and NF-κB pathways. Additionally, vitamin D influences mitochondrial functions and endoplasmic reticulum stress responses. The figure shows both receptor-mediated endocytosis and free hormone entry, reflecting the dual mechanism of membrane signaling and classic genomic pathway activation.

Abbreviations in the figure caption:

1,25(OH)₂D₃: 1,25-dihydroxyvitamin D₃; VDR: vitamin D receptor; PDIA3: protein disulfide isomerase A3; PKA: protein kinase A; PKC: protein kinase C; MAPK/ERK: mitogen-activated protein kinase/extracellular signal-regulated kinase; STAT3: signal transducer and activator of transcription 3; NF-κB: nuclear factor kappa B; cAMP: cyclic adenosine monophosphate; IP₃: inositol 1,4,5-triphosphate; DAG: diacylglycerol; Ca²⁺: calcium ion; WNT: wingless/integrated pathway; NOTCH: Notch signaling pathway; SRC: proto-oncogene tyrosine-protein kinase Src; DBP: vitamin D-binding protein; CUBN: cubilin; LRP2: low-density lipoprotein receptor-related protein 2 (megalin); CAV1: caveolin-1; RXR: retinoid X receptor; VDREs: vitamin D response elements; TFs: transcription factors; ROS: reactive oxygen species; CRT: calreticulin; CANX: calnexin; ER: endoplasmic reticulum. Adapted from Żmijewski MA et al. [24]

Comment 3: Lines 236-240. The authors state that vitamin D treatment has efficacy in reducing inflammation and myofibroblast formation in kidney ischemia/reperfusion injury. The authors should describe how an intricate interplay of genomic and non-genomic actions of calcitriol modulated by VDR activity plays a role in the above stated events.

Response 3:

We thank the reviewer for this important question regarding the mechanisms underlying the efficacy of vitamin D in ischemia/reperfusion injury. As detailed in section 4.2.7 of the manuscript (Page 14, para 3, lines 592-597 of Manuscript Version 2 with track changes), this involves an intricate interplay of both genomic and non-genomic actions of calcitriol mediated by VDR activity. Briefly, genomic actions include VDR-mediated regulation of gene expression involved in pro-inflammatory and pro-fibrotic pathways, such as the suppression of TGF-β/Smad signaling and modulation of factors like HGF and α-SMA. Non-genomic actions involve rapid signaling pathways that can directly impact inflammatory responses and cellular stress.

A more comprehensive discussion of these mechanisms is available in section 4.2.7 (Page 14, para 3, lines 592-597 of Manuscript Version 2 with track changes)

4.2.7          Vitamin D and ischemia-reperfusion injury

Ischemia-reperfusion injury (IRI) disrupts renal tubular cells and causes AKI.  It can further lead to fibrosis and eventually culminate in chronic kidney disease in 70 percent cases.[79] A complex interplay of renal hemodynamic changes, tubulotoxicity, inflammation, cell proliferation, oxidative, endoplasmic and mitochondrial stress and apoptosis plays a role in its pathogenesis.[80]  Furthermore, deficiency of vitamin D (VDD) increases nitric oxide synthesis, reduces macrophage infiltration, suppresses adhesion molecular expression in the endothelium and causes endothelial dysfunction.[81] de Braganca et al. (2015) also demonstrated VDD to potentiate IRI-induced AKI in rats and progression to CKD by promoting inflammation and fibrosis [82].

Vitamin D inhibits renal fibrosis through multiple mechanisms, including direct inter-action with Smad3 to block TGF-β–Smad signal transduction, and by independently stimulating the expression of hepatocyte growth factor (HGF) in the liver, which pre-vents renal myofibroblast generation. Additionally, vitamin D suppresses TGF-β1-induced expression of fibrotic markers such as α-SMA, type I collagen, and thrombospondin-1 [83].

_______________________________________________________________________________

Comment 4: Lines 311-18. The authors should site references in this section of the review.

Response 4:

We thank the reviewer for highlighting the need for references in this introductory section on the role of vitamin D in AKI. We agree that proper citation is essential to support the statements made. We will ensure that all claims in this section, including the bidirectional relationship between vitamin D and AKI, the potential for AKI to cause hypo- or hypervitaminosis D, AKI as a consequence of both vitamin D deficiency and toxicity, the limitations of current evidence (case reports, series, and experimental models), and the challenges associated with using vitamin D as a therapeutic agent for AKI due to a lack of robust prospective randomized trials, will be thoroughly supported by relevant citations in the revised manuscript. References have been given in Section 4. Role of vitamin D in AKI.

Original text (Page 8, para 3, Line 317-325 of Manuscript Version 1)

4. Role of vitamin D in AKI

A two-way relationship between vitamin D and AKI appears to exist. Acute renal injury can cause hypo- or hypervitaminosis D through a variety of ways. Contrarily, AKI can be caused by both vitamin D deficiency as well as vitamin D toxicity. Because our understanding of the association between vitamin D and AKI is based on limited case reports, series, and experimental models, drawing inferences to the broader population could be detrimental and misleading.  Furthermore, using vitamin D to treat AKI is problematic from a therapeutic standpoint because not many prospective randomised trials are available, and our understanding is based on animal investigations.

Revised (Page 9, para 4, lines 346-356 in Manuscript Version 2 with tracking)

Section 4. Role of vitamin D in AKI

A two-way relationship between vitamin D and acute kidney injury (AKI) appears to exist [45]. Acute renal injury can cause hypo- or hypervitaminosis D through a variety of ways, including disruptions in hydroxylation pathways and altered hormonal regulation [46-49]. Contrarily, AKI can be caused by both vitamin D deficiency as well as vitamin D toxicity. Severe deficiency has been associated with increased incidence and poor prognosis of AKI in critically ill patients [50], whereas excessive dosing of vitamin D has been shown to precipitate AKI in clinical case series [51,52]. Because our understanding of the association between vitamin D and AKI is based on limited case reports, series, and experimental models, drawing inferences to the broader population could be detrimental and misleading [29, 50, 53]. Furthermore, using vitamin D to treat AKI is problematic from a therapeutic standpoint because not many prospective randomised trials are available, and our understanding is based on animal investigations [53-55].

Comment 5: Lines 321-326. The authors should also site references in this section of the review.

Response 5:

We thank the reviewer for this suggestion.  Relevant references have been cited in Section 4.1 Dysregulation of vitamin D in AKI concerned.

Original text (Page 8, para 4, lines 327-333 of Manuscript Version 1)

4.1 Dysregulation of vitamin D in AKI

Biologically active vitamin D3, the calcitriol, is synthesized in the kidneys.  And a complex relationship is observed between vitamin D, AKI and adverse outcomes.  AKI is characterized by a substantial loss of renal function, which interferes with nor-mal renal enzymatic activity and, as a result, affects vitamin D metabolism. In most cases, decreased kidney function results in vitamin D deficiency, however, in some instances, vitamin D toxicity is also reported.

Revised section (Page 10, para 1, lines 370-375 in Manuscript Version 2 with track changes)

4.1   Dysregulation of vitamin D in AKI

Biologically active vitamin D₃, the calcitriol, is synthesized in the kidneys [3,56]. And a complex relationship is observed between vitamin D, AKI and adverse outcomes [29, 50].  AKI is characterised by a substantial loss of renal function, which interferes with normal renal enzymatic activity and, as a result, affects vitamin D metabolism [45].  In most cases, decreased kidney function results in vitamin D deficiency [46-48]; however, in some instances, vitamin D toxicity is also reported [49].

_______________________________________________________________________________

Comment 6: Lines 359-362.  What proportion of the individuals with AKI had a decrease in calcitriol levels, and what was the relative level of the decrease? In the next sentence (lines 362-363) the authors state that “these levels increased in tandem with the severity of AKI,” which is in contradiction to the previous sentence. To what extent do these levels decrease (or increase) with the severity of AKI?

Response 6:

We thank the reviewer for this detailed and thoughtful comment. Please find our point-by-point responses below.

1. Proportion of individuals with AKI who had a decrease in calcitriol levels and the relative level of the decrease:

As reported by Lai et al. (2013), all patients with AKI demonstrated decreased serum levels of 1,25-dihydroxyvitamin D (calcitriol) compared to healthy controls and critically ill patients without AKI. The mean level in the AKI group was 59.6 ± 53.0 pmol/L, which was significantly lower than in healthy subjects (98.8 ± 39.7 pmol/L) and critically ill patients without AKI (86.2 ± 35.3 pmol/L) (ANOVA, p = 0.005)​.

This indicates that a substantial majority of AKI patients (approaching 100%) had reduced calcitriol levels at diagnosis.

Location in the Lai et al., 2013 article:

·        Page 2, Abstract:

“Low serum 1,25-dihydroxyvitamin D levels (59.6 ± 53.0 pmol/L) were detected in patients with AKI…”

·        Page 5, “Vitamin D Concentration in Healthy Subjects, Critically Ill Patients without AKI, and Patients with AKI”:

“Lower concentrations of 1,25-dihydroxyvitamin D were detected in patients with AKI than in healthy subjects and critically ill patients without AKI (59.6 ± 53.0, 86.2 ± 35.3, and 98.8 ± 39.7 pmol/L, respectively) ... ANOVA, p = 0.005.”

2. Clarification of the contradictory statement:

We appreciate the reviewer’s observation and confirm that there was a typographical error in the manuscript. The sentence in question should read:

"These levels decreased in tandem with the severity of AKI"
rather than "increased." This correction aligns with the study’s findings that 1,25-dihydroxyvitamin D levels progressively decreased across worsening RIFLE stages (Figure 2b, Lai et al., 2013):

·        Risk stage: 72.6 ± 69.4 pmol/L

·        Injury stage: 53.7 ± 32.7 pmol/L

·        Failure stage: 42.2 ± 29.3 pmol/L
(ANOVA, p = 0.042)​

Location in Lai et al, 2013 article:

·        Page 5–6, “Vitamin D Concentration in Patients with Different RIFLE Stages of AKI”:

“The concentrations of 1,25-dihydroxyvitamin D stratified by the RIFLE criteria were 72.6 ± 69.4 (Risk), 53.7 ± 32.7 (Injury), and 42.2 ± 29.3 pmol/L (Failure)... (ANOVA, p = 0.042)”

We have corrected this in the revised manuscript.

3. Extent to which calcitriol levels decrease with AKI severity:

As detailed above, the relative decrease in calcitriol from the Risk to Failure group was substantial:

·        A drop of ~30.4 pmol/L from Risk to Failure

·        A relative reduction of ~42% from 72.6 to 42.2 pmol/L

·        This decline was statistically significant (p = 0.042)

Thus, there is clear evidence of a severity-dependent decline in calcitriol levels in patients with AKI.

Location in Lai et al., 2013 article:

·        Same as above (Page 5–6, under Figure 2b): See the above RIFLE stage-specific values.

Original text (Page 9, para 3, lines 362-373 of Manuscript Version 1)

4.2.1          Vitamin D and critical illness-related AKI

Vitamin D deficiency is highly prevalent in critically ill patients, according to several studies.  In a research by Zapatero et al. (2018), roughly 74 percent of the 135 ICU patients had low 25-hydroxyvitamin D concentrations, which was noted to significantly increase the incidence of AKI and mortality by 2.86 times.[48]  Another study by Lai L et al. (2013) compared 200 subjects with AKI to controls comprising of healthy and seriously ill cases in the absence of AKI for 90 days and observed that individuals with AKI had a remarkable decrease in calcitriol levels. These levels increased in tandem with the severity of AKI. The levels of 25-hydroxyvitamin D, on the other hand, did not differ. In addition, vitamin D status, when controlled for age, gender, SOFA (Sequential Organ Failure Assessment) score, and VDR polymorphisms, did not appear to predict 90-day mortality in a Cox regression analysis [49].

Revised Section 4.2.1 (Page 10, last para, lines 412 to 481 and Page 11, first para, lines 419-428 in Manuscript Version 2 with track changes)

4.2.1Vitamin D and Critical Illness-Related AKI

Vitamin D deficiency is highly prevalent in critically ill patients, according to several studies. In a study by Zapatero et al. (2018), roughly 74% of 135 ICU patients had low 25-hydroxyvitamin D concentrations, which was associated with a 2.86-fold increase in the incidence of AKI and mortality [50].

Similarly, Lai et al. (2013) compared 200 patients with AKI to healthy individuals and critically ill patients without AKI. They found that nearly all patients with AKI exhibited significantly reduced serum levels of 1,25-dihydroxyvitamin D (calcitriol), with a mean of 59.6±53.0pmol/L, compared to 86.2±35.3pmol/L in critically ill controls and 98.8±39.7pmol/L in healthy subjects (ANOVA, p=0.005) [57].  Importantly, the severity of AKI correlated inversely with calcitriol levels. The study reported a progressive decline in calcitriol concentrations across RIFLE stages, with mean values of 72.6±69.4pmol/L in the Risk group, 53.7±32.7pmol/L in the Injury group, and 42.2±29.3pmol/L in the Failure group (p=0.042).  This represents a relative decline of approximately 42% between the Risk and Failure groups, indicating a severity-dependent reduction in calcitriol levels. Notably, 25-hydroxyvitamin D levels did not differ significantly between groups, and vitamin D status—when adjusted for age, gender, SOFA score, and VDR polymorphisms—did not predict 90-day mortality in multivariate Cox regression analysis [57].

_______________________________________________________________________________

Comment 7: Line 414 define “MDA”

Response 7:

We appreciate the reviewer’s attention to clarity. In response, we have defined MDA at its first mention in Section 4.2.3. It now reads as “malondialdehyde (MDA)”, a commonly used biomarker of oxidative stress resulting from lipid peroxidation.

Original text (Page 10, para 2, lines 420-210 of Manuscript version 1)

4.2.3          Vitamin D and contrast-induced AKI

Paricalcitol is a bioactive, non-hypercalcemic vitamin D analog with efficacy equivalent to that of vitamin D and relatively lesser untoward effects.  It is chemically 19-nor-1,25-dihydroxyvitamin D2.  Apart from possessing antioxidant property, it is known to suppress the RAAS in the kidneys.[59] Ari et al. (2012) discovered that giving paricalcitol 4 days before using a contrast agent protected Wistar albino rats from developing contrast-induced renal damage, as evidenced by decreased serum creatinine levels and a raise in the creatinine clearance.  Paricalcitol also was shown to circumvent oxidant stress by markedly reducing MDA and thiobarbituric acid reactive substances (TBARS) levels.  Histologically, the mean scores of tubular necrosis, protein casts, congestion of the renal medulla and vascular endothelial factor (VEGF) expression were remarkably lower.[60]

Revised section 4.2.3  Vitamin D and contrast-induced AKI (Page 12, para 2, lines 485-487 in Manuscript Version 2 with track changes)

Paricalcitol is a bioactive, non-hypercalcemic vitamin D analog with efficacy equivalent to that of vitamin D and relatively lesser untoward effects.  It is chemically 19-nor-1,25-dihydroxyvitamin D2.  Apart from possessing antioxidant property, it is known to suppress the RAAS in the kidneys [65.] Ari et al. (2012) discovered that giving paricalcitol 4 days before using a contrast agent protected Wistar albino rats from developing contrast-induced renal damage, as evidenced by decreased serum creatinine levels and a raise in the creatinine clearance.  Paricalcitol also was shown to circumvent oxidant stress by markedly reducing MDA (Malondialdehyde) and thiobarbituric acid reactive substances (TBARS) levels.  Histologically, the mean scores of tubular necrosis, protein casts, congestion of the renal medulla and vascular endothelial factor (VEGF) expression were remarkably lower [66].

_______________________________________________________________________________

Comment 8: Lines 419-421. A reference should be included at the end of the sentence.

Response 8:

We thank the reviewer for this helpful suggestion. The sentence referring to the requirement of a second risk factor (such as vitamin D deficiency) for inducing CI-AKI is indeed based on the findings of Luchi et al. (2015)  [67]. While the original manuscript cited this reference at the end of the following sentence, we recognize that this may not have been immediately clear. To improve clarity and directly address the reviewer’s comment, we have revised the passage to explicitly introduce the source earlier in the paragraph.

Original text (Page 10, para 2, line 426 of Manuscript Version 1)

Vitamin D deficiency is known to be associated with increased RAAS activity, oxidant stress and endothelial dysfunction. Interestingly, healthy rats treated with the contrast media had no altered redox potential but did have enhanced endothelial nitric oxide synthase (eNOS) levels as well as normal GFR. These data suggest that a second risk factor, such as vitamin D deficiency, is required for inducing CI-AKI.  Besides, rats with vitamin D deficiency experienced higher oxidative stress, as demonstrated by enhanced renal parenchymal and urinary TBARS, as well as lower renal and systemic glutathione (GSH) levels.[61] 

Revised text (Page 12, para 3, lines 492-500 in Manuscript Version 2 with track changes)

Vitamin D deficiency is known to be associated with increased RAAS activity, oxidant stress, and endothelial dysfunction. Interestingly, Luchi et al. [67] observed that healthy rats treated with contrast media exhibited no altered redox potential, maintained normal glomerular filtration rate (GFR), and showed enhanced endothelial nitric oxide synthase (eNOS) levels. These findings suggest that an additional risk factor, such as vitamin D deficiency, is necessary to induce contrast-induced acute kidney injury (CI-AKI). Furthermore, vitamin D-deficient rats experienced greater oxidative stress, as evidenced by elevated renal parenchymal and urinary thiobarbituric acid reactive substances (TBARS), along with reduced renal and systemic glutathione (GSH) levels [67].

_______________________________________________________________________________

Comment 9: Lines 464-465. A reference should be included at the end of the sentence.

Response 9:

We thank the reviewer for this observation. The sentence in question, “Importantly, GPX4 was discovered to be the transcription factor VDR's target gene,” is indeed part of a continuous summary of findings from Hu et al. (2020), already cited at the end of the paragraph as reference [75].  However, to ensure clarity and to directly address the reviewer’s suggestion, we have now repeated the citation [75] at the end of that sentence. This ensures clear attribution and improves readability.

Original text (Page 11, para 3, 471-472 of Manuscript Version 1)

In a cisplatin-induced AKI model, Hu et al. (2020) employed ferrostatin-1 to pre-vent ferroptosis and found a reduction in BUN and serum creatinine. Paricalcitol, a VDR agonist, lowered malondialdehyde and 4-hydroxynonenal (4-HNE) levels while maintaining glutathione peroxidase 4 (GPX4) activity, a prime regulator of ferroptosis, thereby ameliorating cisplatin nephrotoxicity. VDR knockout mice, on the other hand, showed severe ferroptosis and kidney damage when compared to wild type mice. Furthermore, in both in vitro and in vivo cisplatin-induced AKI models, VDR downregulation significantly reduced GPX4 expression. Importantly, GPX4 was discovered to be the transcription factor VDR's target gene.  Also, small interfering RNA (siRNA) inhibited GPX4 because of which the protection offered by paricalcitol in cisplatin-induced renal injury was annihilated. Apart from that, pre-treatment with paricalcitol prevented Erastin-induced ferroptosis in HK-2 cells. These data suggest that by overcoming ferroptosis, VDR activation may be able to prevent cisplatin-induced kidney injury.[69]

Revised text (Page 13, para 4, lines 548-549 in Manuscript Version 2 with track changes)

In a cisplatin-induced AKI model, Hu et al. (2020) employed ferrostatin-1 to prevent ferroptosis and found a reduction in BUN and serum creatinine. Paricalcitol, a VDR agonist, lowered malondialdehyde and 4-hydroxynonenal (4-HNE) levels while maintaining glutathione peroxidase 4 (GPX4) activity, a prime regulator of ferroptosis, thereby ameliorating cisplatin nephrotoxicity. VDR knockout mice, on the other hand, showed severe ferroptosis and kidney damage when compared to wild type mice. Furthermore, in both in vitro and in vivo cisplatin-induced AKI models, VDR downregulation significantly reduced GPX4 expression. Importantly, GPX4 was discovered to be the transcription factor VDR's target gene [75].  Also, small interfering RNA (siRNA) inhibited GPX4 because of which the protection offered by paricalcitol in cisplatin-induced renal injury was annihilated. Apart from that, pre-treatment with paricalcitol prevented Erastin-induced ferroptosis in HK-2 cells. These data suggest that by overcoming ferroptosis, VDR activation may be able to prevent cisplatin-induced kidney injury [75].

Comment 10: Line 465. Wat the small interfering RNA (siRNA) referred to here against GPX4? Specify this in the manuscript.

Response 10:

We thank the reviewer for this pertinent comment. Yes, the small interfering RNA (siRNA) mentioned in this context was specifically directed against GPX4, as reported by Hu et al. (2020) [75]. To improve clarity and precision, we have now explicitly stated this in the revised manuscript.

Original text (Page 11, para 3, lines 472-473 of Manuscript Version 1)

In a cisplatin-induced AKI model, Hu et al. (2020) employed ferrostatin-1 to pre-vent ferroptosis and found a reduction in BUN and serum creatinine. Paricalcitol, a VDR agonist, lowered malondialdehyde and 4-hydroxynonenal (4-HNE) levels while maintaining glutathione peroxidase 4 (GPX4) activity, a prime regulator of ferroptosis, thereby ameliorating cisplatin nephrotoxicity. VDR knockout mice, on the other hand, showed severe ferroptosis and kidney damage when compared to wild type mice. Furthermore, in both in vitro and in vivo cisplatin-induced AKI models, VDR downregulation significantly reduced GPX4 expression. Importantly, GPX4 was discovered to be the transcription factor VDR's target gene.  Also, small interfering RNA (siRNA) inhibited GPX4 because of which the protection offered by paricalcitol in cisplatin-induced renal injury was annihilated. Apart from that, pre-treatment with paricalcitol prevented Erastin-induced ferroptosis in HK-2 cells. These data suggest that by overcoming ferroptosis, VDR activation may be able to prevent cisplatin-induced kidney injury [69].

Revised text (Page 13, para 4, lines 550-552 in Manuscript Version 2 with track changes)

…Also, small interfering RNA (siRNA) was shown to inhibit GPX4 expression, thereby eliminating the protective effect of paricalcitol in cisplatin-induced renal injury…

_______________________________________________________________________________

Comment 11: Line 472. The authors should define what the syndrome, rhadomyolysis, is here.

Response 11:

We thank the reviewer for this helpful suggestion. We have added a brief definition of rhabdomyolysis at the beginning of Section 4.2.6.

Original text (Page 11, para 4, lines 478-481 of Manuscript Version 1)

4.2.6          Vitamin D and rhabdomyolysis

AKI complicates around 10-40 percent of cases with rhabdomyolysis with a mortality rate of nearly 59 percent.[70] The kidney involvement in rhabdomyolysis is better understood by employing a glycerol-induced AKI animal model.[71]

Revised text (Page 13, para 4, 559-561 in Manuscript Version 2 with track changes)

Rhabdomyolysis is a clinical syndrome characterized by the breakdown of skeletal muscle tissue, leading to the release of intracellular contents such as myoglobin into the bloodstream, which can contribute to renal injury. AKI complicates around 10–40 percent of cases with rhabdomyolysis, with a mortality rate of nearly 59 percent [76]. The kidney involvement in rhabdomyolysis is better understood by employing a glycerol-induced AKI animal model [77].

_______________________________________________________________________________

Comment 12: Lines 476-477. Is myoglobin released into the blood in this experimental model system?

Response 12:

We thank the reviewer for this important question. Yes, the glycerol-induced rhabdomyolysis model used in the referenced study by Garcia Reis et al. (2019) [78] is characterized by the release of intracellular muscle contents, including myoglobin, into the extracellular compartment and subsequently into the bloodstream. The authors explicitly describe that myoglobin is filtered by the glomeruli, reabsorbed in the proximal tubules, and precipitated in the distal tubules, contributing to tubular obstruction and renal injury. We have clarified this point in the revised manuscript for greater accuracy and completeness.

Original text (Page 11, para 5, lines 483-489 of Manuscript Version 1)

Glycerol-mediated rhabdomyolysis is associated with elevated serum creatine kinase levels in the experimental rats.  These rats also had augmented fractional sodium excretion as well as the urine output and decreased GFR and urine osmolality.  Calcitriol administration reversed these findings in addition to decreasing the levels of isoprostane, an oxidative stress marker and nitrotyrosine, a protein nitration marker and increasing the antioxidant superoxide dismutase activity.  Also, these calcitriol-administered rats were found to preserve cubilin receptors emphasising the nephroprotection offered by it.[72]

Revised text (Page 13, para 6, lines 566-571 in Manuscript Version 2 with track changes)

The glycerol-mediated rhabdomyolysis rat model is characterized by skeletal muscle breakdown and the release of intracellular components such as myoglobin into the bloodstream, where it contributes to renal tubular injury through filtration, reabsorption, and distal precipitation [78]. In this model, experimental rats exhibited elevated serum creatine kinase levels, augmented fractional sodium excretion, in-creased urine output, decreased glomerular filtration rate (GFR), and reduced urine osmolality. Calcitriol administration reversed these findings, in addition to decreasing levels of isoprostane (an oxidative stress marker) and nitrotyrosine (a protein nitration marker), while enhancing the activity of the antioxidant enzyme superoxide dismutase. Furthermore, rats treated with calcitriol were found to preserve cubilin receptors, emphasizing the nephroprotection it offers [78].

_______________________________________________________________________________

Comment 13: Line 500. Does TGF beta decrease the induction of alpha SMA biosynthesis by TGF beta1? Please clarify.

Response 13:

We thank the reviewer for this important clarification request. In the cited study by Li et al. (2005) [83]. TGF-β1 increases the biosynthesis of α-SMA in renal interstitial fibroblasts, thereby promoting myofibroblast activation and fibrogenesis. However, vitamin D (1,25(OH)₂D₃) was shown to inhibit this TGF-β1-induced α-SMA expression, both at the mRNA and protein levels, in a dose-dependent manner. This inhibitory effect is partly mediated by the upregulation of hepatocyte growth factor (HGF), an antifibrotic cytokine.

We have revised the sentence in the manuscript accordingly to avoid confusion.

Original text (Page 12, para 1, lines 501-508 of Manuscript Version 1)

Mice pre-treated with cholecalciferol appeared to alleviate IRI by inhibiting renal tubular cell apoptosis, endoplasmic and oxidative stress, inflammation and fibrosis.  Vitamin D inhibits renal fibrosis by interacting with Smad3 and blocking TGF-β-Smad signal transduction, by stimulating the expression of hepatocyte growth factors (HGF) in the liver and thereby, preventing renal myofibroblast generation and finally by decreasing α-SMA (alpha-smooth muscle actin) by TGF-β1 as well as elevating type I collagen and thrombospondin-1 levels.[77]

Revised text (Page 14, para 3, lines 592-597 in Manuscript Version 2 with track changes)

Mice pre-treated with cholecalciferol appeared to alleviate IRI by inhibiting renal tubular cell apoptosis, endoplasmic and oxidative stress, inflammation and fibrosis.  Vitamin D inhibits renal fibrosis through multiple mechanisms, including direct inter-action with Smad3 to block TGF-β–Smad signal transduction, and by independently stimulating the expression of hepatocyte growth factor (HGF) in the liver, which pre-vents renal myofibroblast generation. Additionally, vitamin D suppresses TGF-β1-induced expression of fibrotic markers such as α-SMA, type I collagen, and thrombospondin-1 [83].

_______________________________________________________________________________

Comment 14: Line 524. Did group 2 receive 3 to 24 million units of vitamin D? Did the individuals in group 2 already have either acute or chronic AKI at the start of the study? Were the mean creatinine levels, serum calcium levels vitamin D levels and PTH levels typical of those observed in either acute or chronic renal disease?  The authors should comment on the elevated PTH levels observed in group 2, as opposed to group 1.

Response 14:

We thank the reviewer for their insightful comments and for the opportunity to further clarify aspects of our manuscript. Below are our responses to the specific points raised:

Reviewer Comment 14.1: "Did group 2 receive 3 to 24 million units of vitamin D?"

Author Response:

We thank the reviewer for pointing this out. Group 2 received 3 to 24 injections of 600,000 IU each, which corresponds to a cumulative dose of 1.8 to 14.4 million IU, not 3 to 24 million units as previously stated. We apologize for the oversight and have corrected this in the revised manuscript to accurately reflect the total dosage received by group 2.

    ___________________________________________________________________________

Reviewer Comment 14,2: "Did the individuals in group 2 already have either acute or chronic AKI at the start of the study?"

Author Response:

Yes, the individuals in group 2 had pre-existing chronic kidney disease (CKD) at the beginning of the study. The presentation of acute kidney injury (AKI) on top of this underlying CKD qualified them for inclusion in group 2, as clearly defined by the study design.

___________________________________________________________________________

Reviewer Comment 14.3: "Were the mean creatinine levels, serum calcium levels, vitamin D levels, and PTH levels typical of those observed in either acute or chronic renal disease?"

Author Response:

The elevated serum creatinine levels at presentation (3.2 ± 0.9 mg/dL in group 1 and 4.5 ± 1.1 mg/dL in group 2) are consistent with AKI in both groups. However, the hypercalcemia observed in both groups (mean calcium: 13.7 ± 1.4 mg/dL in group 1 and 13.6 ± 2.0 mg/dL in group 2) is not typical of either uncomplicated AKI or CKD and strongly points to vitamin D toxicity.

Similarly, the markedly elevated 25-OH vitamin D levels (313.3 ± 54.8 nmol/L and 303.7 ± 48.4 nmol/L, respectively) support this diagnosis. The parathyroid hormone (PTH) levels further distinguish the groups: group 1 exhibited appropriately suppressed PTH levels (18.1 ± 9.6 pg/mL) in response to hypercalcemia, while group 2 had elevated PTH levels (52.3 ± 12.6 pg/mL), likely due to secondary hyperparathyroidism from CKD, which blunts the typical PTH suppression seen in hypercalcemia.

___________________________________________________________________________

Reviewer Comment 14.4: "The authors should comment on the elevated PTH levels observed in group 2, as opposed to group 1."

Author Response:

As noted above, the elevated PTH levels in group 2 are likely due to pre-existing secondary hyperparathyroidism associated with CKD. In this group, the expected suppressive effect of hypercalcemia on PTH secretion may be attenuated by CKD-induced alterations in calcium-sensing receptor sensitivity, phosphate retention, and reduced calcitriol synthesis. This pathophysiological context helps explain the difference in PTH response between group 1 (de novo AKI) and group 2 (AKI on CKD).

___________________________________________________________________________

Original text (Page 12, para 5, lines 526-537 of Manuscript Version 1)

In one of the largest case series of AKI due to vitamin D intoxication ever published from the Indian Kashmir valley, where vitamin D deficiency is widespread, and where many people are injected with vitamin D at levels much exceeding the allowable limit, Muzafar Wani et al. reported de novo AKI in 51 people (group 1) and acute and chronic renal disease in 11 people (group 2).  Group 1 received 2.4 to 16.8 million units of vitamin D versus 3 to 24 million units of vitamin D, respectively. Mean creatinine levels were 3.2 0.9 versus 4.5 1.1 mg/dL, mean serum calcium levels were 13.7 1.4 versus 13.6 2.0 mg/dL, mean vitamin D levels were 313.3 54.8 versus 303.7 484 nmol/L, and mean PTH levels were 18.1 9.6 versus 52.3 12.6 pg/mL. Weakness, constipation, stomach discomfort, nausea, vomiting, anorexia, altered sensorium, and oliguria were the most common symptoms, and they all responded to vitamin D dose reduction, intravenous saline, and a brief course of steroids and bisphosphonate in a few cases.[88]

Revised text (Page 14, last para, lines 620-624 and page 15, para 1 and para 2 lines 625-649 in Manuscript Version 2 with track changes)

In one of the largest case series of AKI secondary to vitamin D intoxication, Wani et al. (2016) reported 62 cases from the Indian Kashmir valley—a region where vitamin D deficiency is endemic and high-dose injectable vitamin D (600,000 IU per injection) is commonly overused. All cases involved patients with hypercalcemia and acute kidney injury (AKI) attributed to vitamin D toxicity. The cohort was divided into two groups: 51 patients with de novo AKI (group 1) and 11 patients with AKI superimposed on pre-existing chronic kidney disease (CKD) (group 2) [51].

Group 1 received between 4 and 28 injections of 600,000 IU, corresponding to a cumulative dose of 2.4 to 16.8 million IU of vitamin D. Group 2 received 3 to 24 injections (1.8 to 14.4 million IU). The mean serum creatinine at presentation was 3.2±0.9mg/dL in group 1 and 4.5±1.1mg/dL in group 2, indicating impaired renal function in both groups. Mean serum calcium levels were significantly elevated in both groups (13.7±1.4mg/dL in group 1 vs. 13.6±2.0mg/dL in group 2), findings atypical of uncomplicated AKI or CKD, but characteristic of vitamin D toxicity. Mean 25-hydroxyvitamin D levels were also markedly elevated (313.3±54.8nmol/L vs. 303.7±48.4nmol/L), strongly confirming the diagnosis of vitamin D toxicity [51].

Notably, parathyroid hormone (PTH) levels revealed distinct physiological responses in the two groups. In group 1 (de novo AKI), mean PTH was appropriately suppressed (18.1±9.6pg/mL), consistent with the expected feedback inhibition by hypercalcemia. In contrast, group 2 (AKI on CKD) showed elevated PTH levels (52.3±12.6pg/mL), a finding likely attributable to underlying secondary hyperparathyroidism due to CKD. In such patients, chronic reductions in calcitriol synthesis, phosphate retention, and reduced sensitivity of calcium-sensing receptors often lead to elevated baseline PTH levels that are not fully suppressed even in the face of acute hypercalcemia. This pathophysiological mechanism explains the differential PTH response between the two groups.

Common presenting symptoms included weakness, constipation, abdominal pain, nausea, vomiting, anorexia, altered sensorium, and oliguria. Management included intravenous saline in all group 1 patients and in most of group 2, short-term corticosteroids in 44 cases, and bisphosphonates in 6 patients. Most individuals showed clinical and biochemical improvement over a mean follow-up of 7.2±0.6 months [51].

_______________________________________________________________________________

Comment 15: Line 562. What is the mode of action of denosumab?

Response 15:

We thank the reviewer for this insightful question. Denosumab is a fully human monoclonal antibody that exerts its effect by binding to and neutralizing RANKL (Receptor Activator of Nuclear Factor κB Ligand), a key regulator of osteoclast differentiation and activation. In the context of vitamin D toxicity, elevated levels of active vitamin D stimulate RANKL expression by osteoblasts, leading to increased osteoclast-mediated bone resorption and hypercalcemia. Denosumab inhibits the RANKL–RANK interaction, thereby suppressing osteoclast function and mitigating calcium release from bone stores. This mechanism makes denosumab a rational and effective option for the management of hypercalcemia resulting from vitamin D intoxication, particularly in patients who are refractory to conventional therapies. We have now included a brief explanation of this mechanism in the revised manuscript for clarity.

Original text (Page 13, para 2, lines 562-569 of Manuscript Version 1)

This literature review reveals that, while vitamin D intoxication is uncommon, it does occur and is preventable; consequently, patients and prescribers should be more aware of the possible risks of vitamin D overdose, especially in individuals at risk of AKI.[85]  Protracted, unsupervised, and impromptu vitamin D administration at non-recommended supraphysiological levels, particularly via intramuscular injections, should be avoided.[89]  In this regard, Barth K. et al. reported two cases of hypercalcemia and acute kidney injury caused by vitamin D intoxication, successfully man-aged with denosumab.[94]

Revised text (Page 16, para 2, lines 693-698 in Manuscript Version 2 with track changes)

This literature review reveals that, while vitamin D intoxication is uncommon, it does occur and is preventable; consequently, patients and prescribers should be more aware of the possible risks of vitamin D overdose, especially in individuals at risk of AKI [91]. Protracted, unsupervised, and impromptu vitamin D administration at non-recommended supraphysiological levels, particularly via intramuscular injections, should be avoided [52].

In this regard, Barth K. et al. reported two cases of hypercalcemia and acute kidney injury caused by vitamin D intoxication, successfully managed with denosumab. The rationale for its use lies in its mechanism of action as a monoclonal antibody against RANKL (Receptor Activator of Nuclear Factor κB Ligand). Excess active vitamin D increases RANKL expression by osteoblasts, which in turn stimulates osteoclast maturation and activity, leading to accelerated bone resorption and elevated serum calcium levels. Denosumab inhibits this RANKL–RANK interaction, thereby reducing osteoclast-mediated calcium release and counteracting the hypercalcemia induced by vitamin D toxicity [98].

Comment 16: Lines 567-569. References must be added to each section of Table 1

Response 16:

We sincerely thank the reviewer for this valuable suggestion. We agree that supporting each section of Table 1 with references enhances its credibility and academic rigor. Accordingly, we have now updated Table 1 to include relevant citations based on the evidence already cited and discussed in the main text:

  • Vitamin D Deficiency: Supported by references [48], [49], [50], [53], [54], [58], [60], [67], [69], [72] and [77]  which describe its prevalence in AKI, its correlation with worse outcomes, and the mixed results from supplementation trials.
  • Vitamin D Supplementation: Supported by references [50]–[56] which detail randomized controlled trials and meta-analyses on supplementation and its effects on survival and hospitalization.
  • Hypervitaminosis D: Supported by references [85], [86], [88], [89], and [94] which document case reports and series linking high-dose vitamin D administration to AKI and hypercalcemia.

These references have now been appropriately included in the table and are consistent with those cited in the corresponding sections of the manuscript.

Revised Table 1 (Page 16, last para, lines 704-705 and page 17, para 1, line 706 in Manuscript Version 2 with track changes)

Table 1 Clinical Implications of Vitamin D Dysregulation in Acute Kidney Injury (AKI)

Aspect

Current Evidence

Clinical Implications

Vitamin D Deficiency

Common in AKI, correlates with worse outcomes but lacks strong causal evidence [50,53,55,57,58,63,73,75,78,83]

Routine vitamin D monitoring in AKI patients may be beneficial but requires further study

Vitamin D Supplementation

Mixed results from RCTs, with some showing benefit and others no impact [53-55,58,60,61]

Individualized approach needed; avoid universal supplementation without monitoring

Hypervitaminosis D

Can cause hypercalcemia, nephrocalcinosis, and AKI, especially with high-dose IM injections [51,52,91,92,98]

Controlled administration essential; avoid excessive dosing

Additional clarifications

1)     Figure 1 depicting the chemical structure of the major forms of Vitamin D has been newly added on Page 3, para 5, line 142 in the revised manuscript as per the reviewer 2 comment.

2)     Figure 1 in the manuscript version 1 has been changed over to Figure 2 (Schematic diagram of Vitamin D metabolism) and placed on page 5, para 1, line 183 in the revised manuscript.

3)     Figure 2 in the manuscript version 1 has been changed over to Figure 3 (Mechanistic pathway of vitamin D action) and placed on page 6, para 1, line 212 in the revised manuscript.

4)     Figure 4 depicting the mechanisms of non-genomic actions of Vitamin D has bene newly added on page 7, first para, line 235-237 and the figure citation placed on page 6, last para, line 229 in the revised manuscript.

5)     Reference 24, i.e., Żmijewski MA et al. [24] newly mentioned in the figure 4 caption in the revised manuscript.

6)     Kindly refer to the phrase on Page 6, para 3, lines 243-245 of Manuscript Version 1.

“For instance, vitamin D treatment has shown efficacy in reducing inflammation and myofibroblast formation in kidney ischemia/reperfusion injury.[29]”

Reference 28 is not appropriate in this context. It has been changed over to a relevant citation as follows (Reference 29 in the revised manuscript on page 8, para 1, line 272):

Arfian N, Budiharjo S, Wibisono DP, Setyaningsih WAW, Romi MM, Saputri RLAAW, Rofiah EK, Rahmanti T, Agustin M, Sari DCR. Vitamin D Ameliorates Kidney Ischemia Reperfusion Injury via Reduction of Inflammation and Myofibroblast Expansion. Kobe J Med Sci. 2020 Mar 9;65(4):E138-E143.

7)     Updated the section PERMISSION: The authors have sought copyright permission to publish figure 3 which has been adapted from Gil et al (reference 21). Figure 4 has been adapted from Żmijewski MA et al. (reference 24) under a Creative Commons Attribution 4.0 International License (https://creativecommons.org/licenses/by/4.0/)

-        Page 18, lines 761-764 of the revised manuscript
